# Human mobility on Cancun Island during the Late Postclassic: Intra- and inter-site demographic interactions

**Andrea Cucina**[1☯]*, **Erin Kennedy Thornton**[2☯], **Allan Ortega-Muñoz**[3☯]*

**1** School of Anthropological Sciences, Universidad Autónoma de Yucatán, Mérida, Yucatán, México, **2** Department of Anthropology, Washington State University, Pullman, Washington, United States of America, **3** Physical Anthropology Section, INAH Center Quintana Roo, Chetumal, Quintana Roo, México

☯ These authors contributed equally to this work.
* allan_ortega@inah.gob.mx (AOM); cucina@correo.uady.mx (AC)

**Data Availability Statement:** All the information presented in Tables 1 and 2 can be used to reproduce our results and confirm our conclusions. The included data also provides

## Abstract

Residential mobility in Prehispanic Mesoamerica is of paramount importance in bioarchaeology to determine the "how, where and why" people established biological, political and economic networks. The goal of this paper is to assess the presence of non-local people on the East Coast of the Yucatan Peninsula during the Late Postclassic (AD 1200–1540), and how they might have been perceived by the local Maya people. We analyze the presence, origin and mortuary distribution of 50 individuals based on their dental and bone $^{87}Sr/^{86}Sr$ signatures in the "urban" assemblage constituted by the archaeological sites known as El Rey and San Miguelito on Isla Cancun. Both sites present a strontium ratio "plateau" between 0.7091 and 0.7092, which is considered the local signature. Seven individuals, ranging in age from 5 years old to adulthood, were detected as potentially non-local, and originating from a wide range of regions both near and distant to Isla Cancun. Funerary patterns, burial location, and dietary data do not indicate these people were treated differently from the rest of the population, at least at the moment of death. This suggests that these non-locals might not have been perceived as "foreigners" and that they had integrated into the local community. Such mobility and cultural integration could have motivated by multiple factors, including economic and trade connections, or kinship relationships.

## Introduction

The chronological and cultural Late Postclassic period in the northern Maya lowlands (AD 1200–1540) was characterized by a series of important sociopolitical transformations [1]. During this period, and after the fall of important inland centers like Chichen Itza and Mayapan, the Yucatan Peninsula's eastern territories (i.e., the East Coast) experienced the development of many settlements along the coast [2, 3]. This geographic and demographic expansion along the coast was driven by an array of factors, but the development of expanded Postclassic coastal trading routes was among the most important. Expansion of coastal settlements also implies greater mobility within and between regions and the arrival of foreign people to the East Coast [3:95], with migration occurring at the scale of individuals, families, or whole groups [4–6]. In

comparative 87Sr/86Sr values to support further research.

**Funding:** This study was granted by the project presented by AC, named: Movilidad demográfica en Meso- y Centroamérica en época prehispánica: análisis isotópico y de morfología dental, with the grant number CB-2017-2018-A1-S-10037 from the National Council of Science and Technology of Mexico (Conacyt, in Spanish, https://conacyt.mx/). The funders had no role in study design, data collection and analysis, decision to publish, or preparation of the manuscript. The funders had no role in study design, data collection and analysis, decision to publish, or preparation of the manuscript.

**Competing interests:** The authors have declared that no competing interests exist.

fact, it is in the period following the Classic Maya collapse that Thompson [7] places the arrival into the peninsula of foreign "Mexican" cultures. Opposite to the Classic period (AD 250–900), the Postclassic in this region saw the rise of several walled, fortified towns such as Xcaret and Tulum. Vargas Pacheco [3] hypothesized that this new fortified system meant that the ruling elite might not have been autochthonous.

Previous studies on residential mobility in the peninsula's East Coast [6, 8] indicate that approximately 20% of the population was composed of non-local people during the Late Postclassic. Although there is variation among sites, this percentage is similar to what has been reported previously at other Classic and Postclassic Maya sites [5, 9–11]. Previous studies of Postclassic residential mobility on the East Coast further reveal that most of the non-locals present $^{87}$Sr/$^{86}$Sr values consistent with geographic origins in the northern Maya lowlands and the Mexican and Guatemalan Peten [6], but not places so far away to be consistent with the direct hypothesized arrival of Chontales, Putunes, Toltecas or Mexicas people from the modern-day regions of central Mexico or Tabasco.

Within this socio-political context, and based on Flores Hernández and Pérez Rivas' [1:110] hypothesis that cosmopolitan centers with a population from different regions might have characterized the East Coast during the Postclassic, the direct archaeological and bioarchaeological analysis of Prehispanic human remains is of paramount importance for a better understanding of the extent of the potential "incursion" of foreign people into the region in the Late Postclassic, and more so, if they might have been perceived as "others" by the local Maya people or as an integrating part of the community. With this goal in mind, the present study analyzes the presence, origin and mortuary distribution of people considered as non-locals, based on their $^{87}$Sr/$^{86}$Sr signature, in the "urban" assemblage constituted by the archaeological sites known as El Rey and San Miguelito. These two sites are located on Isla Cancun (Cancun Island, where the modern touristic center lays today), along the Yucatan Peninsula's East Coast. In contrast to other important Late Postclassic sites in the region (e.g., Tulum or Cozumel) where communal burial in ossuaries is more common, the mortuary patterns at El Rey and San Miguelito consist of primary interments. This provides an opportunity to directly compare the burials goods and mortuary treatments of local vs. non-local individuals as potential indicators of different cultural traditions related to the regions of origin. Moreover, the sites of El Rey and San Miguelito were not protected by a walled structure, which suggests that the political elite were not supposed to live there. Therefore, they can be informative of the extent to which these coastal settlements might have been places of attraction for foreign groups not directly related to the political authority or to the socio-political relevance of the site itself, but rather to trading activities and/or other kin-related reasons.

## Postclassic demographic, economic and sociopolitical transitions

In the Yucatan Peninsula's East Coast, four major features characterized the Postclassic period [3, 12]. The first is the intensification of long-distance trading, in particular along the maritime coastal routes. Though it had been present in the peninsula since the Late Classic [13] [9, 10], it is in the Postclassic that it reached its apex, with connections all the way to the Gulf of Honduras and the Gulf of Mexico [13–15]. The second is the process of militarization that, in the East Coast, saw the rise of fortified towns like Tulum, Xcaret and Ichpaatun among others (see Vargas Pacheco [16] for a list of walled towns). These fortifications required a huge economic and political effort by the ruling elite [17:165] [3:30]. Militarism by the theocratic elite emerged in the region as a response to increased social complexity, population density, and labor division, and the concentration of economic sources that benefited those holding the power [3:31]. According to Vargas Pacheco [3], the elite ruling the walled towns did not belong to the

same ethnic group as the locals; they were foreigners who established their power in those towns.

Militarism has been more thoroughly analyzed in Yucatan inland territories (i.e., Mayapan) [18:140], but the topic has received much less consideration on the East Coast. After AD 1250, and the fall of Mayapan, militarism increased, and alliances were formed under a confederate political organization [3]. By the time the Spaniards arrived on the peninsula, they found it was divided into 19 chiefdoms including Ecab, which encompassed the northern East Coast [3] including Isla Cancun. López Austin and López Luján [19] indicate that the chiefdom's Maya governors during the Late Postclassic were the so called *halach uinic*, residing in the chiefdom's main town, while the other centers were supervised by a lower ranking *batabo'ob*, who answered to the *halach uinic*.

The third feature of the Postclassic is the demographic expansion and movement of people towards the coast. Such intra- and inter-regional residential mobility has already been documented since Late Classic times in the northern lowlands [5, 6, 9] and along the East Coast itself [6, 8, 20]. As mentioned above, the social and cultural impact of foreign people had been reported by Thompson [7:144] already a century after the so-called "Maya collapse". He called it the Mexican intrusion, based on the appearance of a different architectural style referred to the Chontales groups from Tabasco, the ones Thompson [7:147] named Itzaes. Names like Putunes, Toltecas, Mexicas, Chontales, Itzaes have been frequently assigned to these hypothetical invading groups during the Postclassic. While integrating into local Maya society, these groups are proposed to have introduced new sociocultural, political, and economic models [14, 19: 273], which influenced local architecture, language, and warfare [21]. However, Roys' and Thompson's proposed foreign "invasion" has been recently criticized by Cobos [22], who cites a lack of strong archaeological evidence to support their historic interpretations.

The fourth feature, which is a consequence of the demographic expansion, is the rise of many settlements along the coast. Coastal locations on the eastern side of the peninsula are not the most appropriate places for intensive agriculture suitable for sustaining large populations [3]. This means that settlers had to rely on marine resources and coastal trading for their subsistence [3]. It would not be unexpected therefore that the many small and large centers along the coast, and their people, were involved in trading networks reaching territories beyond the borders of the Maya cultural region. Though trading is an important push factor for residential mobility, Arnauld et al. [4] (see also Cucina et al. [5]) stress that other social, political and kinship factors promoted mobility in the past.

## Study sites

Isla Cancun (Cancun Island) is a narrow strip of land extending 21 km north-to-south and 400 meters east-to-west along the east coast of Mexico's Yucatan Peninsula [23:95]. Today, Isla Cancun harbors the highly touristic "*Zona Hotelera*" of the modern town of Cancun, Quintana Roo, Mexico, which transformed over the last forty year from a small coconut farm village to a major tourist destination. Evidence from the archaeological sites of Koxolnah and El Conchero indicate that human settlement on Isla Cancun dates back to the Late Preclassic (300 BC–AD 250) [23: 99–100] [24] [25:22], but the Isla's precolonial population did not peak until centuries later in the Late Postclassic (AD 1200–1540) [23]. The island's demographic growth, however, was curtailed by the arrival of the Spanish in the mid-16th century AD when early colonial documents record swift depopulation and sites abandonment.

The sites of El Rey and San Miguelito were established during the Early Postclassic (AD 900–1200) at the island's point of maximum width. Based on the number of structures and

settlement patterns, they are considered the main and most densely populated Late Post-classic (AD 1200–1540) communities on the island when they reached their apex in population size and spatial extent [23]. The original Maya names for El Rey and San Miguelito are not known, but according to Vargas Pacheco the two sites might instead represent two different neighborhoods of a single community known as Nizuc (August and Alice Le Plongeon referred to the community as Nizucte during their visit to the archaeological remains in 1877 –[26]:11]). The currently recognized sites of El Rey and San Miguelito are in fact less than one mile apart (Fig 1), and it is unknown whether they were more contiguous in Prehispanic times since looting and the touristic development have heavily affected the area [2]. Despite the relatively small size of these two sites and the apparent lack of a political elite, Andrews [2] considered them as macro-zones, i.e., densely populated zones, based on the number and characteristics of the architectural structures. In other words, regardless of whether they were two contiguous sites or one larger site, they were the largest and most densely populated human settlement(s) in the context of Isla Cancun.

The site of El Rey is located 2.5 km. north of Punta Nizuc and extends for 520 meters north-to-south and 70 meters east-west. El Rey (the King) was given this name because of an anthropomorphic head recovered at the site [23, 27]. Archaeological explorations in the 1970s identified two site sectors, named El Rey (like the site itself) and Pinturas (paintings), with the latter sector named after now disappeared mural painted stuccos. All structures in both sectors were built in the Mexican or Eastern Coast style; the walls are either sloped or straight with a cornice molding and the stairs have nicely (or squarely) finished balustrades among other characteristics. Such stylistic approach characterized the whole East Coast in the Late Postclassic (see [12]:88–89] [23]:107]). Its "El Rey" sector contains 47 structures with the primary structures (structures 1 to 7) located in the sector's center (Fig 2). Nine platforms with small altars in their front side delineate a roadway running north-to-south. Columnated structures with a central staircase and an altar are reminiscent of the major Postclassic capital of Mayapan in the northern Yucatan Peninsula.

Located south of the El Rey sector, the "Pinturas" sector contains 27 structures including platform mounds, civic and ceremonial buildings, palaces, and altars. As in the El Rey sector, structures delineate a roadway through the sector's center. The so-called "palaces" in the Pinturas sector are similar to other sites along the East Coast including Tulum, San Gervasio, and San Miguelito, but such architectural features are absent at the major inland capitals of Chichen Itza and Mayapan.

San Miguelito is located a mile north of El Rey. Despite evidence of occupation in the Early Classic [23], this site also reached its apex during the Late Postclassic [28]. Some 40 structures have been reported along a stretch of land 600m long (Fig 3), but the full extent of the site is unknown since many structures may have been destroyed during construction of the *Zona Hotelera* [28]. Five clusters of structures have been described [23, 28]. Similar to Pinturas, four of them contain palaces (one also has a pyramid-like structure), platforms with colonnades, and residential structures. Only one–the northernmost cluster—does not contain palaces, and instead consists of a platform with staircase and colonnade, and five residential structures with staircases at the front and interior columns [23, 28]. Chultunes (artificial underground cave-like structures) provided storage of drinking water at the site.

Given the two sites' location along the coast, the Postclassic population relied on marine resources [23, 28, 29]. However, subsistence and luxury goods (jade, obsidian, gold, copper, among others) were obtained through short- and long-distance trading [23, 28], evidence that stresses the role of Isla Cancun in the maritime coastal trade routes, which connected the East Coast to other parts of Mesoamerica and, likely, Central America.

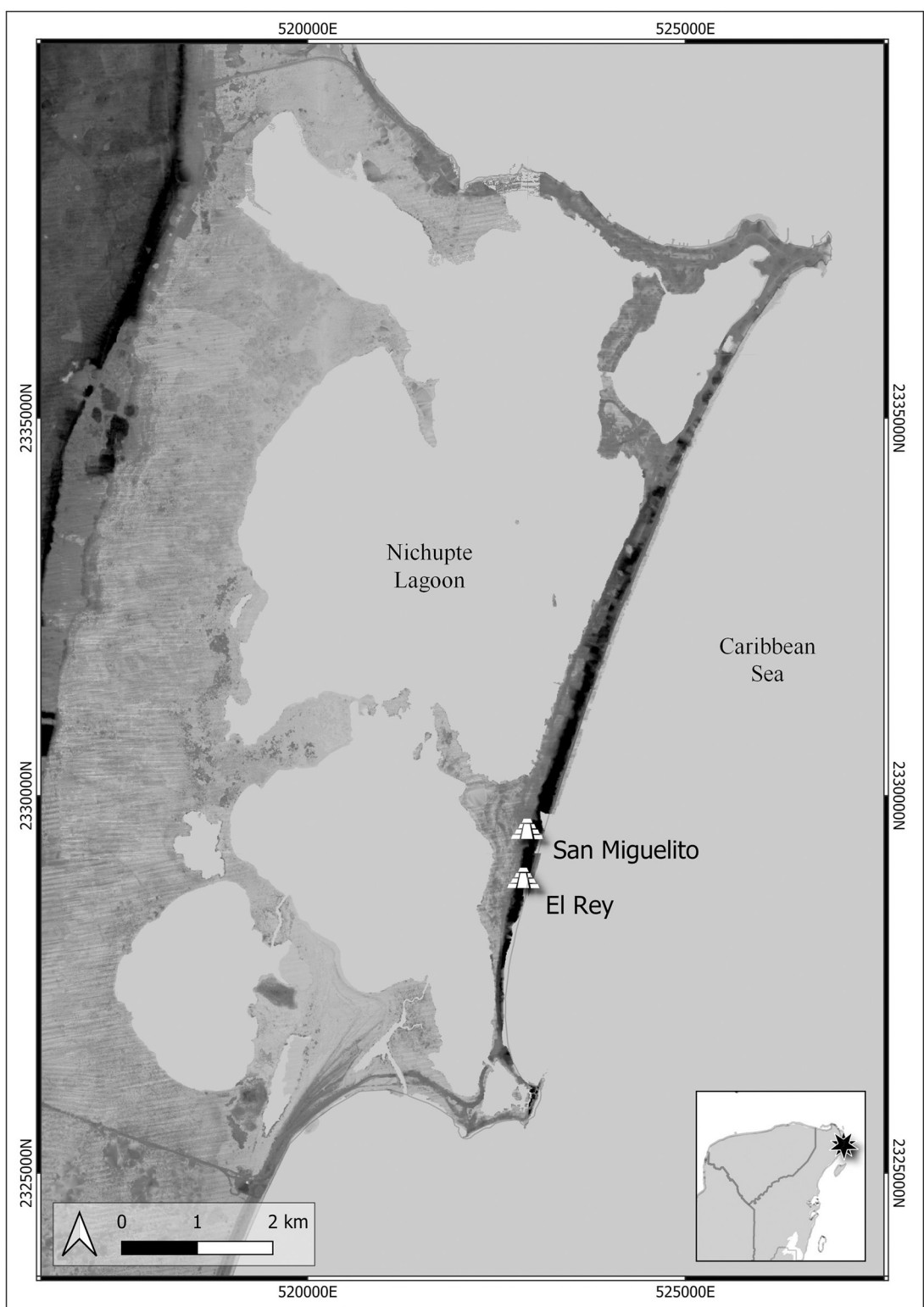

**Fig 1. Map of Isla Cancun showing the position of El Rey and San Miguelito (Five-meter high-resolution LiDAR–INEGI; elaborated under a CC BY license, with permission from Ashuni E. Romero Butrón, copyright 2023).**

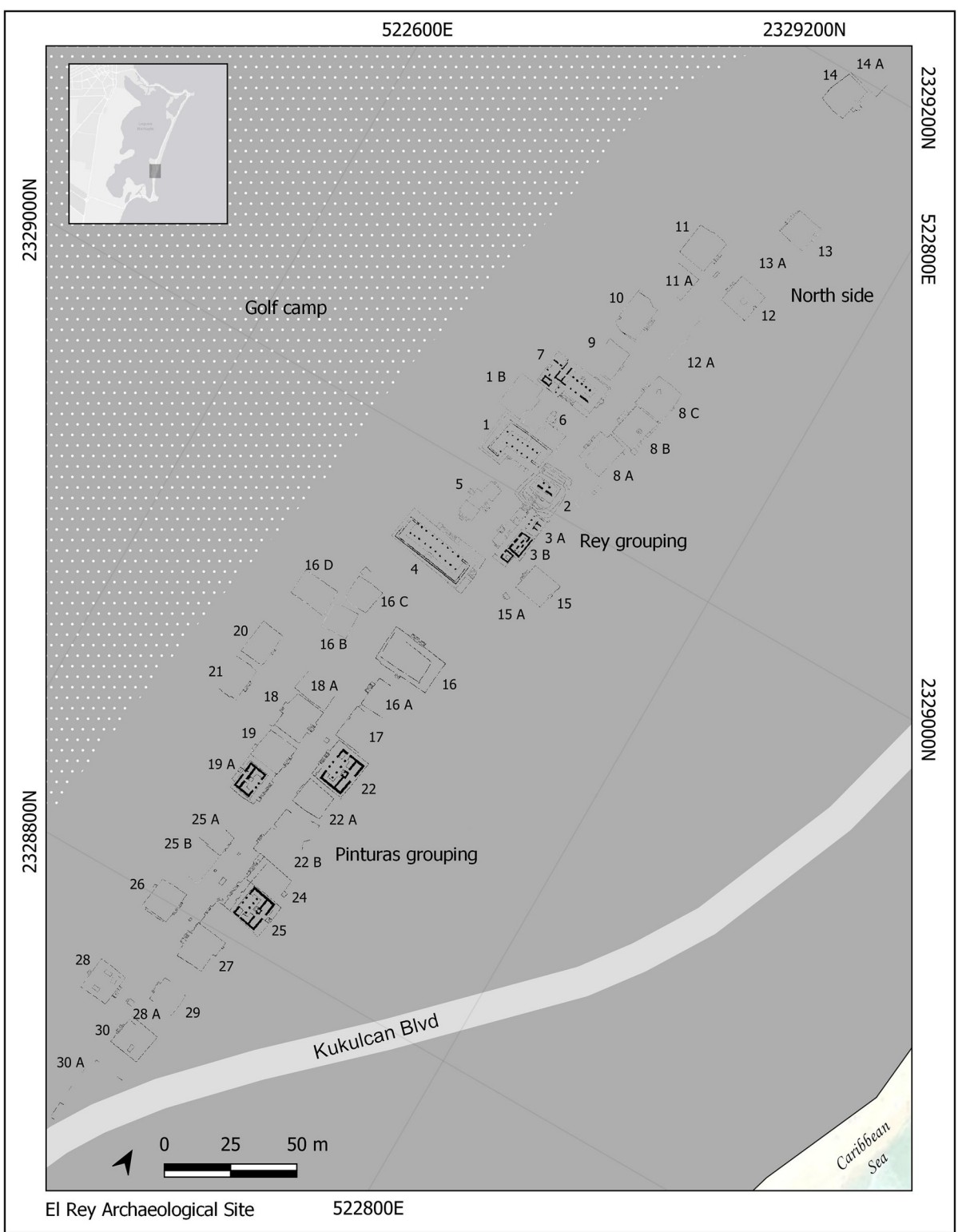

**Fig 2. Map of the site of El Rey, showing the distribution of its structures.** Drawing by Ashuni E. Romero Butrón under a CC BY license, copyright 2023.

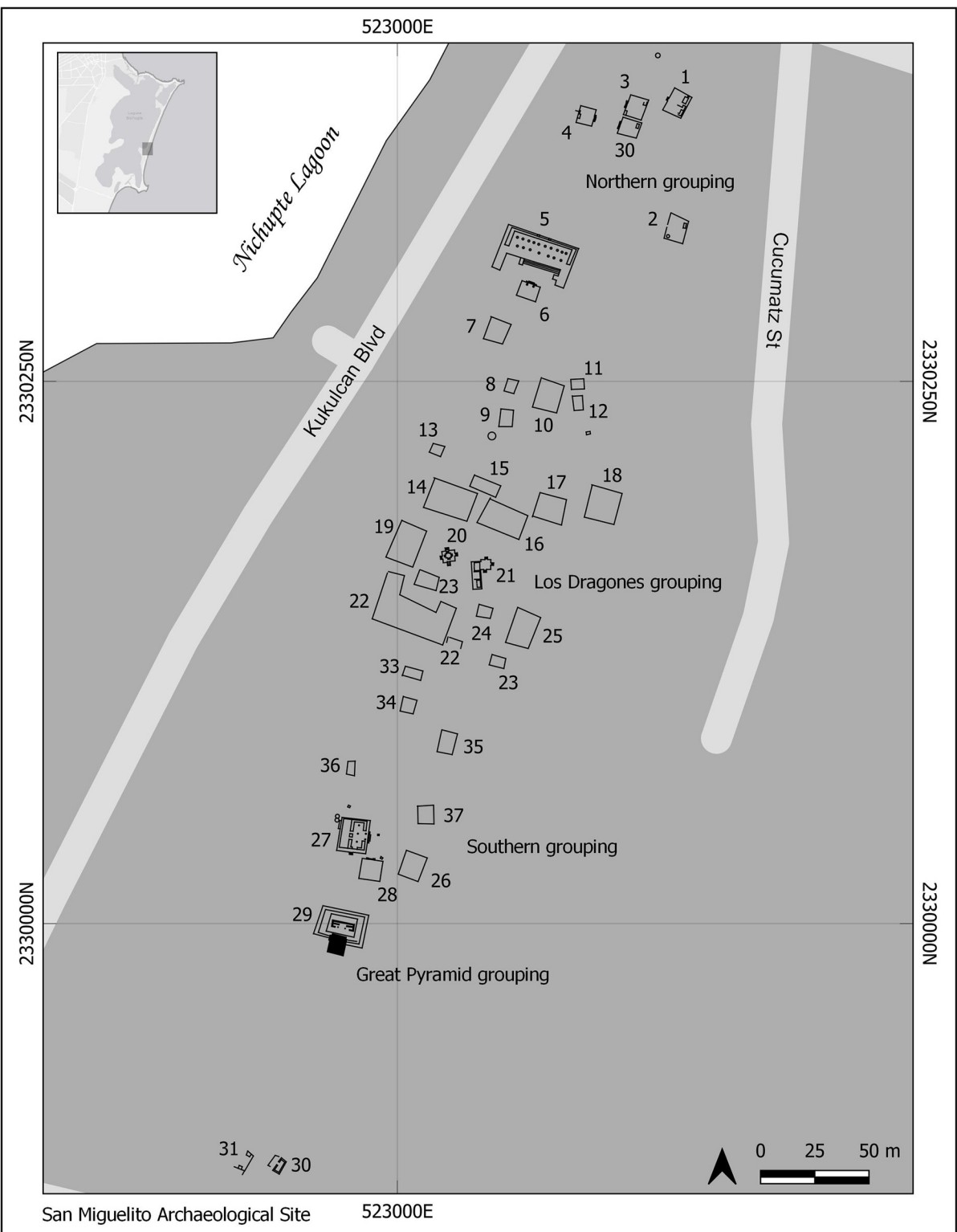

**Fig 3. Map of San Miguelito.** Reprinted from Elizalde [28] under a CC BY license, with permission from Sandra Elizalde, original copyright 2014. Redrawn and modified by Ashuni E. Romero Butrón under a CC BY license, copyright 2023.

## Concepts of $^{87}sr/^{86}sr$ analysis

Strontium isotope analysis is widely used to reconstruct patterns of ancient mobility and migration, based on the fact that strontium isotope ratios ($^{87}Sr/^{86}Sr$) vary across the landscape with the underlying geology. In geological substrates, $^{87}Sr/^{86}Sr$ varies according to a rock's age and original $^{87}Rb/^{87}Sr$ composition since $^{87}Sr$ derives from the radioactive decay of rubidium ($^{87}Rb$). The half-life of $^{87}Rb$, however, is 48.8 billion years, meaning that although the $^{87}Sr/^{86}Sr$ of rocks changes over time, this occurs at a timescale irrelevant to archaeological investigations. Within the Maya cultural area, regions of distinct geology varying in $^{87}Sr/^{86}Sr$ include the northern (~0.7080–0.7092) and southern (~0.7073–07085) lowlands, volcanic highlands and Pacific coast (~0.7038–0.7050), Maya Mountains (>0.7100), and the highly variable metamorphic province (0.7040–0.7202) [30–35] (Fig 4).

Strontium isotope ratios measured in archaeological skeletal remains can be used to infer an individual's region of origin because as humans and other animals feed, the local $^{87}Sr/^{86}Sr$ of the plants and animals consumed is recorded in their skeletal tissues when strontium (Sr) substitutes for calcium (Ca) during bone and tooth mineralization [36–39]. Because human dental enamel does not remodel, it represents the geochemical signature of the place in which the individual was living when the enamel formed. Because the crown of the permanent first molar forms between birth and 3 years of age [40], and that of premolars between 2 and 4 years of age [41:151], those teeth represent the environment the individual was born in and spent his/her early years of life. On the contrary, bone remodels throughout the whole lifetime,

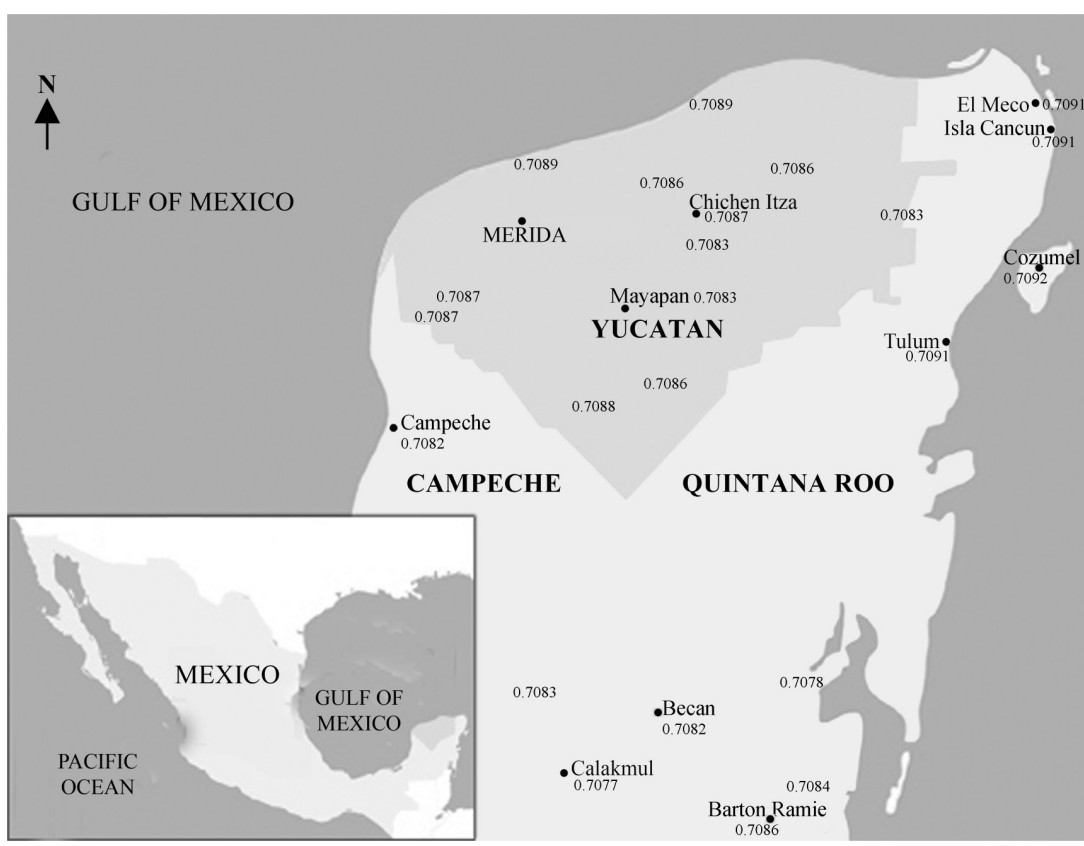

**Fig 4. Map of strontium database in the Maya area.** Drawing by A. Cucina using database from Ortega-Muñoz et al. [20].

with a turnover that occurs at a pace of approximately 10–15 years in adult life [37:413]. The ratio of $^{87}Sr/^{86}Sr$ in human tooth enamel has long been used to detect as non-locals those individuals whose signature does not match the local geochemical signature, and who moved into the place where they were eventually interred sometime during life [38, 42, 43]. The analysis of strontium ratios in bone, instead, allows the assessment of whether non-local individuals moved in during their last decade of life or before it. Local strontium signatures are commonly assessed based on $^{87}Sr/^{86}Sr$ recorded from geological materials, water, plants, or small animals with limited home ranges (e.g., rodents, armadillos, land snails) [31, 42, 44, 45]. Small animals are often the preferred proxy for bioavailable strontium incorporated into human tissues since they average local $^{87}Sr/^{86}Sr$ over their range [36, 43]. Local values and potential places of origin also can be assessed based on published $^{87}Sr/^{86}Sr$ maps for Mesoamerica (e.g., [31, 32, 45, 46]) (Fig 4).

## Materials and methods

The human skeletal remains analyzed in the present study are all archaeological specimens recovered from archaeological contexts in Quintana Roo, Mexico. All human archaeological remains are protected by the Mexican Federal Legislation (*Ley Federal sobre Monumentos y Zonas Arqueológicos, Artísticos e Históricos, 1972*). Specifically, human and cultural archaeological artifacts are managed and protected by the Instituto Nacional de Antropología e Historia (INAH–National Institute of Anthropology and History), which is the Mexican Federal Institution in charge of archaeological materials. All the analyses performed in this study were carried out after receiving the authorization and consent from the Mexican Archaeological Council (authorization n. 401.1S.3-2021/042), INAH´s highest level authority, which supervises the ethical approach to studying human archaeological skeletal remains and approves their use for research purposes. The human burial remains from San Miguelito and El Rey are kept in the Centro INAH Quintana Roo under the custody of Dr. Allan Ortega-Muñoz. The samples are part of the Mexican National Heritage.

Fifty-three human skeletons from El Rey were recovered from eleven structures, rubble deposits and one plaza during the 1975–1978 field seasons conducted by the Mexican National Institute of Anthropology and History (INAH) and the National Autonomous University of Mexico (UNAM). In 2006, three more primary interments were recovered from a sandy deposit [47]. The human osteological collection from San Miguelito was recovered during a salvage intervention by the INAH in 2011–2012 [28], during which 14 of the 40 structures were explored and restored.

Dental enamel from 20 individuals from San Miguelito and 30 from El Rey were analyzed for $^{87}Sr/^{86}Sr$ to assess past human movement after early childhood (Tables 1 and 2). $^{87}Sr/^{86}Sr$ was also analyzed in cortical bone from seven individuals from El Rey and one individual from San Miguelito to determine residence location during the last several years of life [48] (Tables 1 and 2). Intra-individual comparisons of $^{87}Sr/^{86}Sr$ in tooth enamel and bone can thus indicate more recent post-childhood movement. Only distilled water was used during the cleaning process to remove dirt from the skeletal remains to avoid chemical contamination. The restoration process follows the standard procedure like application of water-based glue to joint fragments of bone that fractured during excavation process. However, the bone samples selected for analyses had been separated and did not undergo any restauration procedure to avoid chemical contamination. The skeletal sample comprises individuals of both sexes, aged between 2 and 60 years. Of the 30 individuals from El Rey, seven were from ritual-religious structures, nine from ritual-religious/domestic structures, seven from purely domestic structures, and seven were from rubble, sandy or unidentified deposits. Ten of the individuals from El Rey were

**Table 1. List of individuals analyzed from San Miguelito by sex, age, $^{87}$Sr/$^{86}$Sr, and context.** Rows in italics report results from bone (N.D.–Not Defined).

| Burial | Sex | Age (years) | Element* | Structure | Context | $^{87}$Sr/$^{86}$Sr | Origin | Place |
|--------|-----|-------------|----------|-----------|---------|---------------------|--------|-------|
| 26 | N.D. | 5 | M$^1$ | 1 | Domestic | 0.70909 | Potentially non-local | Nearby northern lowlands or nearby Isla Cancun |
| 29 | N.D. | 5 | M$^1$ | 4 | Domestic | 0.70805 | Non-local | Central Peten |
| 2 | Male | Adult | M$^1$ | 1 | Domestic | 0.70917 | Local | |
| 13 | N.D. | 2 | M$^1$ | 1 | Domestic | 0.70915 | Local | |
| 22 | N.D. | 2 | M$^1$ | 1 | Domestic | 0.70917 | Local | |
| 4 | N.D. | 2 | M$^1$ | 3 | Domestic | 0.70911 | Local | |
| 5 | N.D. | 6 | M$^1$ | 3 | Domestic | 0.70916 | Local | |
| 6 | Male | Adult | M$^1$ | 3 | Domestic | 0.70917 | Local | |
| 7 | N.D. | 2 | M$^1$ | 3 | Domestic | 0.70911 | Local | |
| 12b | N.D. | 3 | M$^1$ | 3 | Domestic | 0.70918 | Local | |
| 14 | N.D. | 4 | M$^1$ | 3 | Domestic | 0.70913 | Local | |
| 15 | Male | 35–60 | M$^1$ | 3 | Domestic | 0.70917 | Local | |
| 18 | Male | 30–34 | M$^1$ | 3 | Domestic | 0.70915 | Local | |
| *18* | *Male* | *30–34* | *Femur* | *3* | *Domestic* | *0.70918* | *Local* | |
| 19 | N.D. | 3 | M$^1$ | 3 | Domestic | 0.70916 | Local | |
| 23 | N.D. | 2 | M$^1$ | 3 | Domestic | 0.70917 | Local | |
| 25 | N.D. | 3 | M$_1$ | 3 | Domestic | 0.70920 | Local | |
| 30 | Male | 25–30 | M | 4 | Domestic | 0.70919 | Local | |
| 37 | N.D. | Subadult | M | 6 | Ritual-religious | 0.70922 | Local | |
| 41 | Female | 20–24 | M | 5 | Domestic | 0.70920 | Local | |
| 42 | Female | Adult | M | Pavilion 2 | Domestic | 0.70922 | Local | |

*M$_1$ First molar lower; M$^1$ First molar upper; M First molar; M$_2$ Second molar lower; M$^2$ Second molar upper; M$_3$ Third molar lower; M$^3$ Third molar upper

recovered from the Pinturas sector with most of them (n = 9) originating from ritual-religious or ritual-religious/domestic structures. At San Miguelito, with only one exception, all individuals were recovered from domestic (residential) structures.

Initial measurements of enamel $^{87}$Sr/$^{86}$Sr from El Rey were carried out at the Geochronology and Isotope Geochemistry Laboratory (GIGL) (Dept. of Geological Sciences, University of North Carolina-Chapel Hill) under the supervision of T. Douglas Price and his team (University of Wisconsin-Madison) [8]. Further analyses of El Rey's dental enamel, the whole set of human bones, as well as the enamel from San Miguelito were performed at the Washington State University Radiogenic Isotope and Geochronology Laboratory (RIGL) under the supervision of Erin Thornton and Jeff Vervoort (RIGL Director). Twelve samples that had been previously analyzed at the GIGL were replicated successfully at the RIGL, demonstrating full comparability of results between laboratories.

At RIGL, $^{87}$Sr/$^{86}$Sr was measured using a ThermoFinnigan Neptune MC-ICPMS. Instrument accuracy was monitored and confirmed through replicate measurements of the strontium standard NBS-987. Samples were prepared and analyzed in a ULPA-filtered class 1000 clean lab to prevent contamination from exogenous Sr. Prior to analysis, archaeological skeletal samples were abraded using a dental drill fitted with a carbide bit to remove dirt, debris and any adhering dentin. After mechanical cleaning, samples were pretreated in a 5% acetic acid solution to remove post-depositional contaminants. Following pretreatment, samples were hot-digested in nitric acid (HNO$_3$ optima) and loaded into cation exchange columns packed with Eichrom strontium-selective resin to isolate strontium from other ions.

**Table 2. List of individuals analyzed from El Rey (P stands for Pinturas) by sex, age and $^{87}Sr/^{86}Sr$ values, use of the structure and potential place of origin.** Rows in italics report results from bone (N.D.–Not Defined).

| Burial | Sex | Age (years) | Element* | Structure | Context | $^{87}Sr/^{86}Sr$ | Origin | Place |
|--------|-----|-------------|----------|-----------|---------|-------------------|--------|-------|
| 6 | Male | 28–32 | $M^2$ | Rubble | N.D. | 0.70903 | Non-Local | Nearby northern lowlands |
| 7 | Male | 28–32 | $M_1$ | 2 | Ritual-religious | 0.70881 | Non-local | Northern coast of Yucatán |
| *7* | *Male* | *28–32* | *Femur* | *2* | *Ritual-religious* | *0.70920* | *Local* | |
| 28a | N.D. | >30 | $M_1$ | 5 | Ritual-religious | 0.70806 | Non-local | Southern lowlands |
| *28a* | *N.D.* | *>30* | *Humerus* | *5* | *Ritual-religious* | *0.70912* | *Local* | |
| 23–1 | Female | >30 | $M_1$ | 7 | Ritual-religious / domestic | 0.70852 | Non-local | Northern coast of Belize or northern Peten |
| *23–1* | *Female* | *>30* | *Ulna* | *7* | *Ritual-religious / domestic* | *0.70920* | *Local* | |
| 32 | Female | Adult | $P^4$ | 9 | Domestic | 0.70903 | Non-local | Nearby northern lowlands |
| *32* | *Female* | *Adult* | *Femur* | *9* | *Domestic* | *0.70917* | *Local* | |
| 36 (P) | Male | 36 | $M^1$ | 1 | Ritual-religious / domestic | 0.70915 | Local | |
| 37 (P) | Male | 30–35 | $M_1$ | 1 | Ritual-religious / domestic | 0.70914 | Local | |
| 38(P) | Female | n.d. | $M_1$ | 1 | Ritual-religious / domestic | 0.70916 | Local | |
| 39 (P) | Female | 22–24 | $M_1$ | 1 | Ritual-religious / domestic | 0.70914 | Local | |
| 40 (P) | Male | 35–40 | $M_3$ | 1 | Ritual-religious / domestic | 0.70917 | Local | |
| 42 (P) | Female | 30–35 | $M_3$ | 2 | Ritual-religious | 0.70916 | Local | |
| 43 (P) | Male | Adult | $M^1$ | 2 | Ritual-religious | 0.70917 | Local | |
| 50 | Male | 45–50 | $M^1$ | 2 | Ritual-religious | 0.70915 | Local | |
| 23–3 | Male | 25–30 | $M_1$ | 3 | Ritual-religious / domestic | 0.70917 | Local | |
| 45 (P) | Female | 28–32 | M | 3 | Ritual-religious | 0.70916 | Local | |
| 46 (P) | Male | 28–32 | $M^1$ | 3 | Ritual-religious | 0.70916 | Local | |
| 23–2 | Female | 16–18 | $M_2$ | 7 | Ritual-religious / domestic | 0.70917 | Local | |
| 27 | Female | 33–35 | $M_1$ | 7 | Ritual-religious / domestic | 0.70916 | Local | |
| 14 | N.D. | >30 | $M_1$ | 8 b | Domestic | 0.7092 | Local | |
| 18 | Female | >30 | $M_1$ | 8 b | Domestic | 0.7092 | Local | |
| *18* | *Female* | *>30* | *Fibula* | *8 b* | *Domestic* | *0.70919* | *Local* | |
| 19 | Female | 33–35 | $M_1$ | 8 b | Domestic | 0.7092 | Local | |
| 21 | Male | 28–32 | $M_1$ | 8 b | Domestic | 0.7092 | Local | |
| 22 | Female | 22–24 | $M_1$ | 8 b | Domestic | 0.7092 | Local | |
| 31(P) | N.D. | Subadult | M | 16 | Domestic | 0.70912 | Local | |
| 2A | Male | >30 | $M^1$ | Sand | No structure | 0.70911 | Local | |
| *2A* | *Male* | *>30* | *Tibia* | *Sand* | *No structure* | *0.70919* | *Local* | |
| 3 | Male | >30 | $M_1$ | Rubble | N.D. | 0.70917 | Local | |
| *3* | *Male* | *>30* | *Tibia* | *Rubble* | *N.D.* | *0.70921* | *Local* | |
| 3 | Male | 30 | $M_1$ | Sand | N.D. | 0.70917 | Local | |
| 10 | Male | 30–45 | $M_2$ | Rubble | N.D. | 0.7092 | Local | |
| 11 | Male | 28–32 | $M_2$ | Rubble | N.D. | 0.7092 | Local | |
| 12 | Female | >30 | $M_1$ | Rubble | N.D. | 0.70915 | Local | |
| *12* | *Female* | *>30* | *Femur* | *Rubble* | *N.D.* | *0.7092* | *Local* | |
| Faunal[1] | | | | | | 0.70920 | | |

*$M_1$ First molar lower; $M^1$ First molar upper; M First molar; $M_2$ Second molar lower; $M^2$ Second molar upper; $M_3$ Third molar lower; $M^3$ Third molar upper; [1]Faunal: Mandible of opossum

To distinguish non-local individuals, the local $^{87}Sr/^{86}Sr$ range for Isla Cancun was established using limited faunal baseline data (i.e., archaeological opossum [*Didelphis* sp.] $^{87}Sr/^{86}Sr$ = 0.7092) and supplemented by previously published values from Mesoamerica (e.g., [31, 32] [45]). Statistical inter-quartile range comparisons of human $^{87}Sr/^{86}Sr$ values were also

conducted using the SPSS 15 package as a complementary approach to identifying non-locals (see [49]).

## Results

Determining the proper baseline is of paramount importance to detect "locals" versus "non-locals". Tables 1 and 2, and Figs 5 and 6 show the isotopic values for every individual analyzed by site, as well as for one animal remain from San Miguelito. Both sites present a narrow range of most individuals between 0.7091 and 0.7092. This range is consistent with the faunal value as well as with the ranges detected in other coastal sites close to Isla Cancun (i.e., El Meco, just north of Cancun, and Tulum) [8] (see Fig 6, which includes El Meco). Hernández-Terrones et al. (2021) [50] analyzed several water sources in the region close to Cancun/Puerto Morelos and found that $^{87}Sr/^{86}Sr$ ratios varied between 0.7080 (one case) and 0.7092, with an average of 0.7088. However, all the locations tested by the authors were some 16 to 22 miles inland from Isla Cancun, and no direct information was provided for the area immediately surrounding the lagoon where the sites of Isla Cancun were built. Hernández-Terrones et al.'s [50] ratios may be suggestive of a potential baseline ranging from 0.7088 and 0.7092 for Isla Cancun and surrounding region. However, if this were true, and people from 16 miles and farther were moving from inland sites to the coast and vice versa, then we should expect to find continuity

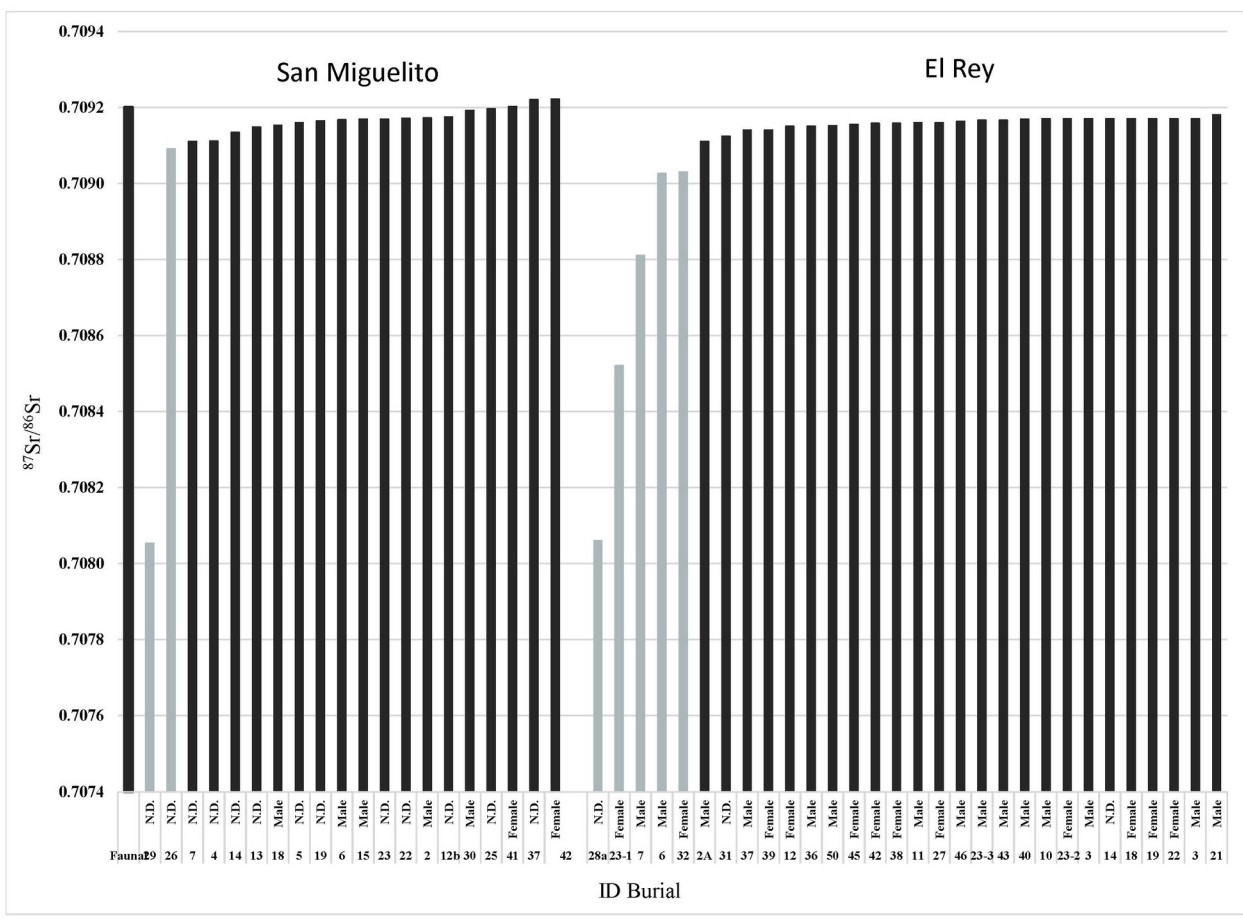

**Fig 5. $^{87}Sr/^{86}Sr$ values from dental enamel San Miguelito and El Rey.** In grey, individuals non-locals.

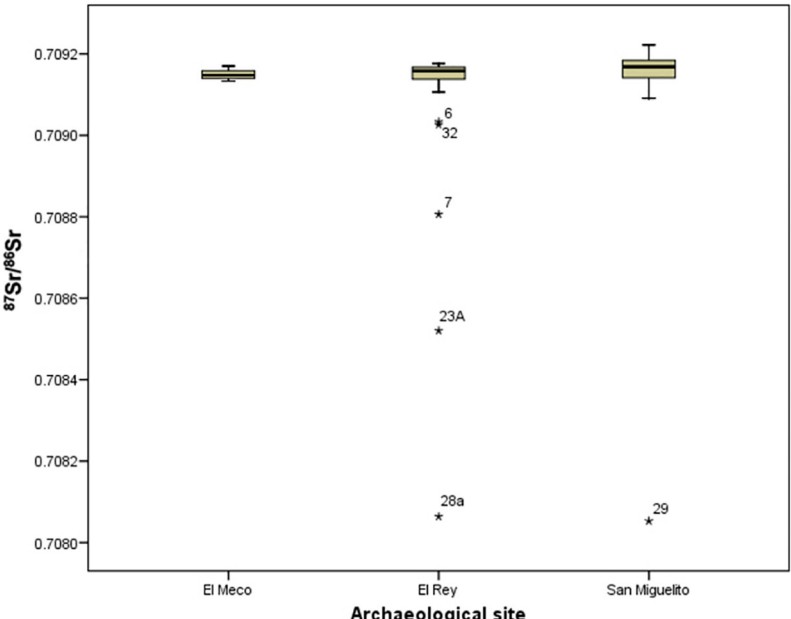

**Fig 6. Box-plots of enamel $^{87}$Sr/$^{86}$Sr isotopic variation at San Miguelito (right), El Rey (center) and El Meco (left)** [6, 8].

of Sr ratios from 0.7088 to 0.7092 in the skeletal collections; people showing Sr ratios lower than 0.7088 therefore should be considered as "non locals". Yet, as we can appreciate from Tables 1 and 2 and Figs 5 and 6, of the 50 people analyzed (20 from San Miguelito and 30 from El Rey), only one individual shows Sr ratio of 0.7088, and two of 0.70903. The strong consistency of Sr ratios above 0.7091 (as well as for El Meco and Tulum) [8] suggests that those people on the very edge of the coast were not drinking water from places located miles inland but were instead eating foods and obtaining drinking water with strontium isotope ratios closer that of modern sea water ($^{87}$Sr/$^{86}$Sr = 0.7092). Therefore, given that 44 out of 50 people (88%) present Sr ratios above 0.7091, and because of the paucity of individuals below that threshold, in particular those within 0.7088 and 0.7091, we consider that local people along the narrow strip of land on the northern part of the East Coast of the peninsula can be characterized by a narrow local $^{87}$Sr/$^{86}$Sr range of 0.7091± 0.00003 (Table 3).

More in detail, at San Miguelito, only one individual, a five-year-old child (Burial 29), yielded a non-local $^{87}$Sr/$^{86}$Sr value of 0.70805, indicating that the individual potentially migrated from Petén, Guatemala or the more distant Gulf Coast. Another five years-old child (Burial 26) presents a ratio of 0.70909, which is a shade below the local range. This child may either be a local individual laying at the lower edge of the local $^{87}$Sr/$^{86}$Sr range, or he/she may have originated from a location nearby Isla Cancun.

In contrast, five individuals at El Rey (Burials 6, 7, 23–1, 28a and 32) present $^{87}$Sr/$^{86}$Sr ratios that fall below the estimated local range, and that are clearly non-local. Potential places of origin, based on available databases [46], are the southern Maya lowlands (e.g., Tikal region) (Burial 28a), northern coast of Belize or northern Peten (Burial 23–1), northern coast of the Yucatan Peninsula (Burial 7), and some place(s) nearby Isla Cancun in the northern Maya lowlands (Burials 6 and 32). The female individual from Burial 32 was originally considered to be non-local [8] based on δ$^{18}$O measured in her tooth enamel.

**Table 3. Descriptive statistics of isotope ratios at San Miguelito and El Rey, with and without potential non-locals.**

| | | San Miguelito | San Miguelito[1] | El Rey | El Rey[1] | Both sites[1] | Both sites[2] |
|---|---|---|---|---|---|---|---|
| | N | 20 | 18 | 30 | 25 | 53 | 8 |
| | Mean | 0.70911 | 0.70917 | 0.70908 | 0.70916 | 0.70916 | 0.70918 |
| | St. Dev. | 0.00025 | 0.00003 | 0.00023 | 0.00002 | 0.00003 | 0.00003 |
| | Minimum | 0.70805 | 0.70911 | 0.70806 | 0.70911 | 0.70911 | 0.70912 |
| | Maximum | 0.70922 | 0.70922 | 0.70918 | 0.70918 | 0.70922 | 0.70921 |
| Q1 | 25 | 0.70914 | 0.70915 | 0.70913 | 0.70915 | 0.70915 | 0.70917 |
| Q2 | 50 | 0.70917 | 0.70917 | 0.70916 | 0.70916 | 0.70917 | 0.70919 |
| Q3 | 75 | 0.70919 | 0.70919 | 0.70917 | 0.70917 | 0.70918 | 0.70920 |
| IQR = Q3-Q1 | | 0.00005 | 0.00004 | 0.00003 | 0.00002 | 0.00003 | 0.00003 |
| Inferior | Q1-k*IQR | 0.70906 | 0.70909 | 0.70908 | 0.70913 | 0.70911 | 0.70912 |
| Superior | Q3-k*IQR | 0.70911 | 0.70913 | 0.70912 | 0.70914 | 0.70914 | 0.70915 |
| k | 1.5 | | | | | | |

[1] Without non-locals;

[2] Only $^{87}$Sr/$^{86}$Sr bone values

Descriptive statistics and interquartile analyses (Table 3) show that San Miguelito and El Rey present very similar values when potential non-locals are excluded from the count. The two collections overlap completely, with practically the same values (Fig 6).

By looking in detail to the values of local individuals from Tables 1 and 2, we note that Burials 4 and 7 from San Miguelito, and Burial 2A from El Rey (all with an $^{87}$Sr/$^{86}$Sr = 0.70911) are very close to the lower limit of the range for local individuals. Given the relative homogeneity of $^{87}$Sr/$^{86}$Sr ratios along the East Coast of the peninsula (see [6, 8]), these individuals might well have been born at the site or in places nearby, like Tulum [8], whose lower range of variability is very similar to that of Isla Cancun. If the latter were the case, Burials 4 and 7 from San Miguelito, who died both at the age of 2 years, moved into Isla Cancun shortly after being born. Burial 4 presents a parallelepiped form of cranial deformation. Such shape was documented by Tiesler [51] as having originated in the region of the Gulf of Mexico and "imported" to the Yucatan Peninsula in the Postclassic. However, there is no doubt that the individual in Burial 4 was not born in such a remote place from where he/she died. Burial 2A, instead, is a >30 years old male who, in case of having not born in Isla Cancun, moved in long before his death, as the $^{87}$Sr/$^{86}$Sr ratio from his bones indicates.

As for Burial 2A, the isotopic evidence of when individuals moved in during their lifetimes indicates that all the non-locals that were analyzed for $^{87}$Sr/$^{86}$Sr ratio in bones spent the last decade of life in Isla Cancun (Fig 7). The same can be said for local individuals, who were born at the sites of El Rey and San Miguelito, who appear to have spent their last decade at the sites since $^{87}$Sr/$^{86}$Sr measured in both their tooth enamel and bone match local values. In all but one cases, $^{87}$Sr/$^{86}$Sr in bone is slightly higher than its equivalent in dental enamel. However, even though it might be indicative of a long-lasting presence of each of those individuals at Isla Cancun before death, it must be taken into consideration that diagenesis might have affected the chemical signature in the bones and artificially shifted them closer to the $^{87}$Sr/$^{86}$Sr of the surrounding depositional environment. Price and colleagues [52: 592] show that $^{87}$Sr/$^{86}$Sr values of bones samples of sailors of the voyages of Columbus to America, buried in the La Isabela's cemetery in Santo Domingo, are generally homogenous (mean: 07092± 0.00002), indicating that these samples are contaminated by local strontium isotopes. The bone samples from the present study have similar $^{87}$Sr/$^{86}$Sr ratios and variability, with a mean of 0.70918±

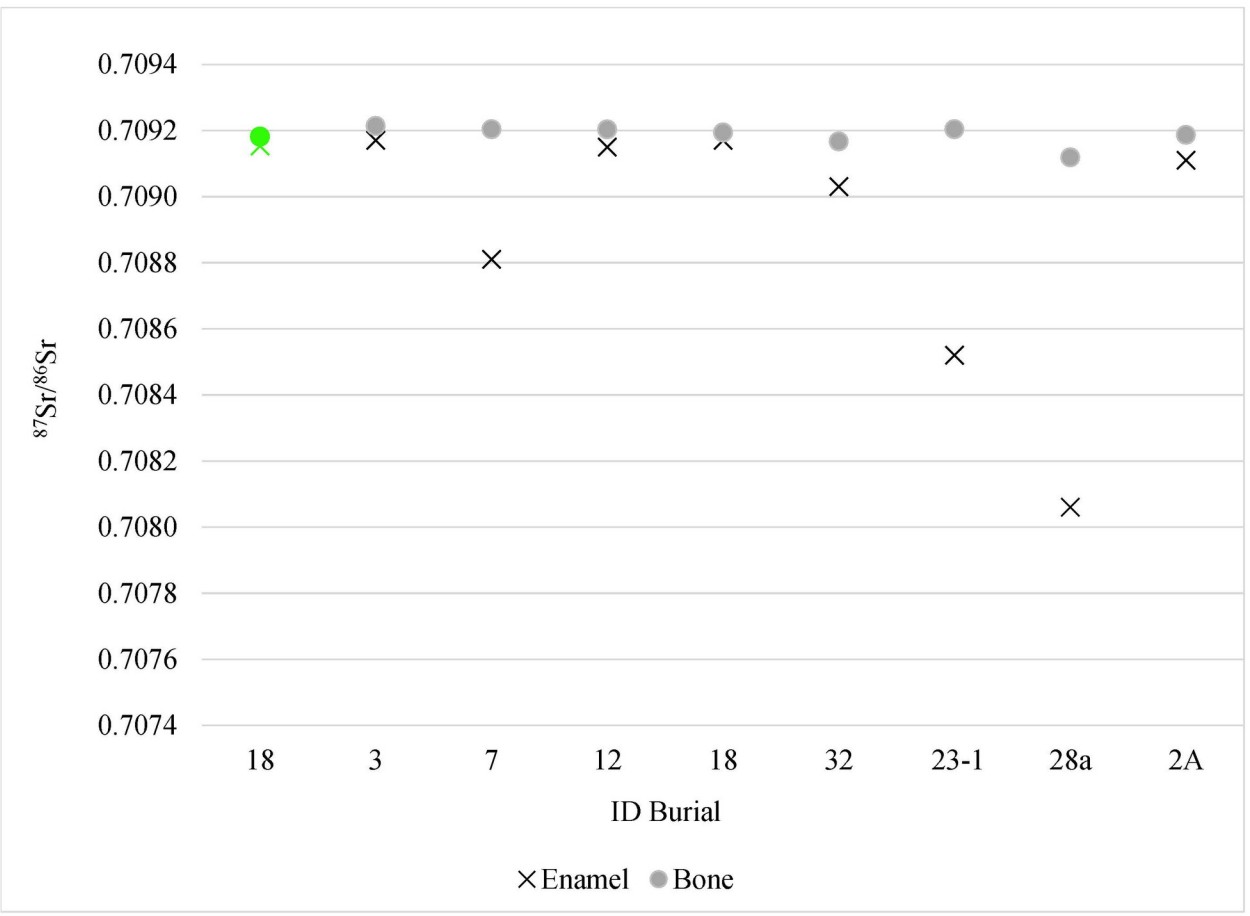

**Fig 7. Paired $^{87}Sr/^{86}Sr$ ratios of dental enamel (cross) and bone (circle) of nine individuals from Isla Cancun.** The individual from San Miguelito is the first from the left (Male, Burial 18) painted in green; El Rey is represented by all the others (in grey).

0.00003 (Table 3), which raises doubts on the validity of Sr data from bony remains because of potential diagenetic contamination. Therefore, the issue of time spent at the site should be considered with much caution, and should be explored further in future investigations.

## Discussion

The $^{87}Sr/^{86}Sr$ ratios in dental enamel shows variability, allowing to detect non-locals in Cancun Island; on the other hand, the same cannot be said for the $^{87}Sr/^{86}Sr$ values obtained from bony remains, which show a strong homogeneity. While contamination during the cleaning process can be ruled out, we cannot completely rule out that such homogeneity in bones might result from diagenesis. Establishing a coherent baseline for strontium isotopes in bones and teeth long the Easy Coast of the Peninsula of Yucatan should represent the focus of future research. Given the inherent doubts on the validity of $^{87}Sr/^{86}Sr$ ratios in bone, a conservative approach must be taken, so that the issue related to the time in life when those individuals arrived at Cancun Island cannot be discussed in this paper.

El Rey and San Miguelito, or Nizuc if they were one single community, were important centers in the Late Postclassic [3], but they do not represent a walled, or fortified community. Despite the community's importance, the lack of a defensive wall and architectural features at

the site indicate that they were not the center of a political organization, like Tulum. They were part of the Ecab chiefdom and were likely supervised by a *batabo'ob* [19].

In this perspective, Burial 7 from the El Rey sector stands out. It is the only burial from the site's single small pyramidal, religious-ritual structure, and it was accompanied by the largest number of associated funerary objects (ceramics, copper axes, deer's hunts, jade and more) [53:108]. Among the associated funerary objects, we recognize items that are clearly allochthonous to the northern Maya lowlands, like copper and jade. This indicates that individual in Burial 7 might have been of higher social and/or economic status than the others. Moreover, the presence of a copper axe suggests that this individual might indeed have been a *batabo'ob* (which means the one of the axes in Maya) [19]. If this were the case, it is interestingly to note that this individual originated in the northern coast of the peninsula, and not in Isla Cancun.

The Ecab chiefdom encompassed the northeastern part of the modern state of Quintana Roo [54]. According to Flores Hernández and Pérez Rivas [1] the town of Ecab, from which the chiefdom is named, and which lays in the northeastern-most corner of the peninsula, was one of five main centers governing the chiefdom. $^{87}$Sr/$^{86}$Sr measured in water samples from the Yalahau region in the northern part of the Ecab province range from 0.707888 to 0.709046 [55]. Given this range of variation, it is very difficult to locate the specific area(s) from which the individual in Burial 7 might have come from. Nonetheless, the fact that a non-local might have possessed such higher status supports the concept that the chiefdom "central" power controlled the region and that the towns in Isla Cancun were not independent settlements [1] [3]. However, this is in contrast with Okoshi Harada [56] who claimed that the Ecab province was instead a series of small political entities independent from one another and linked by commercial and/or religious ties.

Except for the richly attired Burial 7, recovered in a ritual-religious structure, no clear pattern can be highlighted for the other non-local individuals from an architectonic perspective; they were recovered from different structures, rubble, and sand deposits. Nevertheless, the kind of structure itself is irrelevant in relation to the place of burials of non-locals, because two came from religious-ritual structures (including Burial 7), one from a religious-ritual/domestic structure, one from a domestic context and one from rubble (Fig 8). At the same time, the two non-locals (one potentially so) from San Miguelito were recovered from domestic compounds (Fig 9).

Even though it is worthy to note that at El Rey, all five non-local individuals were recovered from the El Rey sector of the site (25%, 5/20), and that all ten individuals from the Pinturas sector were local, there does not seem to be a clear pattern of interment for them. Also being interred inside or outside structures does not seem to be an issue related to the dichotomy of local vs non-local origin. Among the non-locals, only Burial 6 (from the El Rey sector of El Rey) was interred in rubble outside a structure. All the other non-local individuals were buried inside structures. This individual (Burial 6), however, was not alone, because six other individuals (all locals) were from sand and rubble graves outside structures. Ortega-Muñoz and Ramos [53] consider burial outside of structures as representative of the last phase of occupation of the site, when the original structures had lost their initial ritual-religious status, and had been transformed into domestic, residential ones. However, the lack of more detailed chronological evidence within the Late Postclassic phase leaves this interpretation at the level of conjecture.

The lack of a "compound" specifically reserved for non-locals, like the Oaxaca Barrio in Teotihuacan [57, 58], suggests, at least from the mortuary perspective, that non-locals may not have been perceived as "foreigners" and that they had integrated into the local community [21]. In this perspective, of those detected as non-locals (from both El Rey and San Miguelito) only three originated from different and more distant territories, but all within the Maya

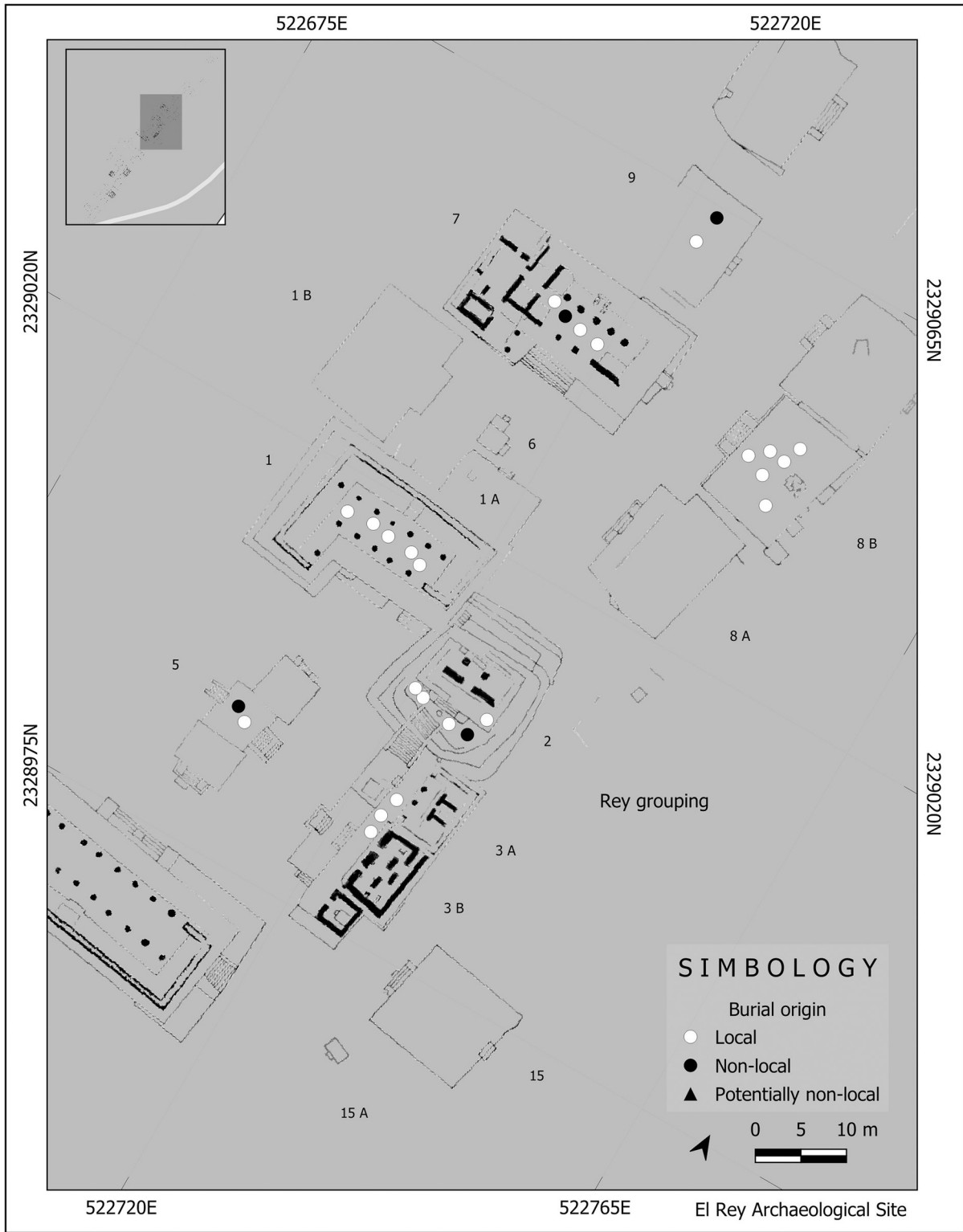

**Fig 8. Location by structure of local and non-local individuals from the archaeological site of El Rey.** Drawing by Ashuni E. Romero Butrón under a CC BY license, copyright 2023.

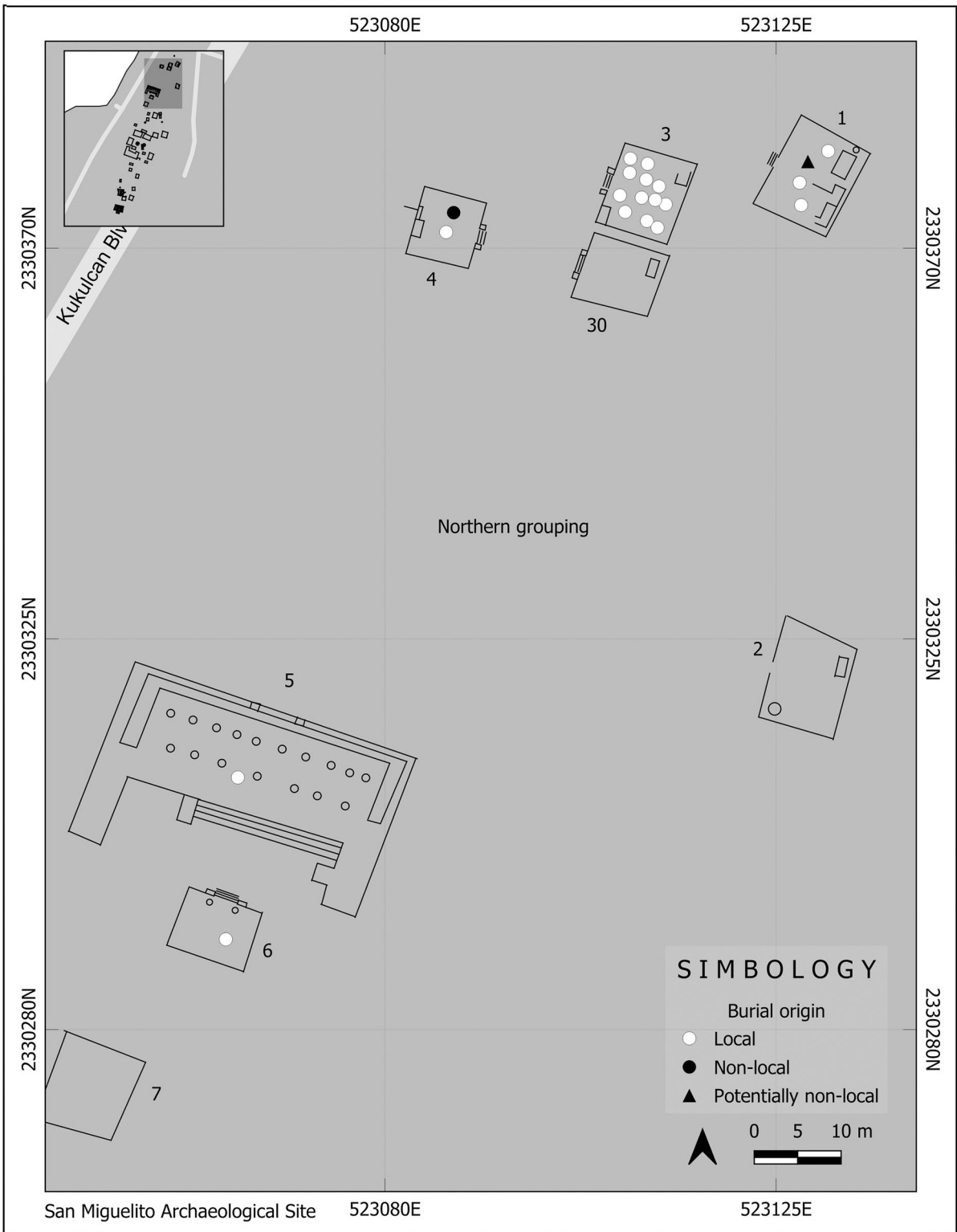

**Fig 9. Location by structures of local and non-local individuals from the archaeological site of San Miguelito.** Modified and redrawn by Ashuni E. Romero Butrón (under a CC BY license, copyright 2023) from Elizalde [28], original copyright 2014.

realm. The others were from nearby areas in the northern Maya lowlands. Scherer et al. [21] point out that the Maya population was not culturally homogeneous. The ancient Maya might have considered as "foreigners" those Mayas from very distant regions, or those not speaking a Mayan language [21]. Definitely, and this is not unexpected (see [6]), Maya from different communities, lineages, or classes that originated from the same region (like the Ecab province, for example) could hardly be seen as foreigners [21:166]. Cultural concept of shared ethnicity might have been linked to their relationship with the ancestors instead of the biological lineage, which would have allowed non-locals to be perceived as locals or cultural "insiders" [21:162].

Cultural and physical integration seems to be supported also by the analysis of dietary isotopes. Duke [29] reports homogeneity in dietary intake based on $\delta^{13}C$, and a retrospective analysis of her data shows no differences between locals and non-locals. After all, as Freiwald [30:205] (see also [6]) stated, non-locals who became integrated into local communities eventually ate the same food, used the same tools (but see Burial 7 from El Rey), and were interred in the same places as the locals. In these cases, only their strontium isotope signatures would identify them as non-locals.

One issue that will deserve further investigation is the apparent lack of non-locals interred in the Pintura sector of El Rey. Cucina and Ortega-Muñoz [6] and Ortega-Muñoz et al. [8] showed that also at the Late Postclassic site of El Meco, a religious center along the coast in front of Isla Mujeres [1], all the people analyzed for $^{87}Sr/^{86}Sr$ fell within the local range. Such lack can simply be due to sample size or to the places from which human remains were recovered. However, given the relatively small size of Pinturas, which was investigated in its entirety, location where archaeological investigations were carried out does not seemingly apply and other factors should eventually be explored.

Similar to what was reported in previous studies from the Classic and Postclassic period northern Maya lowlands' realm [5, 6, 8, 11, 20, 30, 59, 60], both El Rey and San Miguelito show that twelve to fourteen percent of people originated from some place—close or far—away from town. At El Rey, they are all adults (who seemingly moved in several years before death). At San Miguelito, instead, both non-locals (one a potential non-local from a very close-by territory) are infants, which means that their displacement had been decided by someone else (likely their parents). Reasons for residential mobility are multiple, and involve economic reasons, trade connections, and kinship relationships [61]. Cucina and Ortega-Muñoz [6] state that the presence of non-local subadults in the skeletal collections is indicative that families (and not just adult individuals) were moving around. At the same time, the evidence of non-local biocultural markers in local infants, like parallelepiped cranial deformation, which was not the norm for the region [51], strengthens the notion that cultural markers could have been introduced from some distant regions, and applied to local newborns as a form of cultural heritage of the place the infants' parent originated from, such as the case of the infant from Burial 4 from San Miguelito.

Last, similarities in distribution of Sr ratios at San Miguelito and El Rey does not contribute directly with relevant information onto whether these archaeological sites were two separate political entities in Late Postclassic, or just neighborhoods of one larger city. Further, multidisciplinary studies will have to be performed to answer that question in the future.

## Conclusions

In conclusion, El Rey and San Miguelito (or Nizuc if the two sites were indeed one town) was not the place of residence of the political elite, but they were a subordinate center linked to one of the region's main towns. The presence of non-locals falls within the range of variability already highlighted in the Maya realm, which suggests that also in Late Postclassic times

residential mobility was as intense as in previous periods. More so, three (plus one more if we consider the second potential infant from San Miguelito) of the non-locals were from nearby territories in the northern Maya lowland, and only three people were from regions at some distance from the East Coast, but still within the limits of the Maya lowlands. No one came from very distant regions that may suggest a direct immigration of Mexicas, Putunes, Chontales or Itzaes people into Isla Cancun. Mortuary data and grave locations suggest that local and non-local individuals were given similar mortuary treatments, suggesting that migrants integrated into the site's social fabric and were likely perceived as locals by the autochthonous population.

Obviously, one of the limits of strontium isotope analysis is that it cannot detect second-generation "immigrants", so other techniques will be necessary in the future to solve this question. The presence of a potential *batabo'ob* from northern territories in the Ecab region suggests that the chiefdom's main centers were controlling the chiefdom's "minor" towns. Last, residential mobility, for whatever reason it did occur, did not involve only adults but (likely) whole families, including infants and children suggesting a wide array of reason behind people's mobility.

## Supporting information

**S1 Checklist. *PLOS ONE* clinical studies checklist.**
(DOCX)

## Acknowledgments

The authors acknowledge Kimberly Sheets (Washington State University, Dept. of Anthropology and Dr. Jeff Vervoort (Director, Washington State University Radiogenic Isotope and Geochronology Lab) who were critical to completing the isotopic analyses.

## Author Contributions

**Conceptualization:** Andrea Cucina, Allan Ortega-Muñoz.

**Formal analysis:** Andrea Cucina, Allan Ortega-Muñoz.

**Funding acquisition:** Andrea Cucina.

**Investigation:** Andrea Cucina, Erin Kennedy Thornton, Allan Ortega-Muñoz.

**Methodology:** Andrea Cucina, Erin Kennedy Thornton, Allan Ortega-Muñoz.

**Supervision:** Allan Ortega-Muñoz.

**Writing – original draft:** Andrea Cucina, Allan Ortega-Muñoz.

**Writing – review & editing:** Andrea Cucina, Erin Kennedy Thornton, Allan Ortega-Muñoz.

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
