## [Decision Letter · Decision Letter 0]

1 Jun 2023

PONE-D-23-08900Human mobility on Cancun Island during the Late Postclassic: intra- and inter-site demographic interactionsPLOS ONE

Dear Dr. Ortega-Muñoz,

Thank you for submitting your manuscript to PLOS ONE. After careful consideration, we feel that it has merit but does not fully meet PLOS ONE’s publication criteria as it currently stands. Therefore, we invite you to submit a revised version of the manuscript that addresses the points raised during the review process.

We look forward to receiving your revised manuscript.

Kind regards,

Marco Milella

Academic Editor

PLOS ONE

Journal Requirements:

5. We note that Figures 1-4 in your submission contain [map/satellite] images which may be copyrighted. All PLOS content is published under the Creative Commons Attribution License (CC BY 4.0), which means that the manuscript, images, and Supporting Information files will be freely available online, and any third party is permitted to access, download, copy, distribute, and use these materials in any way, even commercially, with proper attribution. For these reasons, we cannot publish previously copyrighted maps or satellite images created using proprietary data, such as Google software (Google Maps, Street View, and Earth). For more information, see our copyright guidelines: http://journals.plos.org/plosone/s/licenses-and-copyright.

a. You may seek permission from the original copyright holder of Figures 1-4 to publish the content specifically under the CC BY 4.0 license.  

Additional Editor Comments:

We now received the reports from both reviewers.

As you will see, they agree about the scientific interest and overall quality of your work.

They do however recommend that some minor issues be addressed (this especially refers to Reviewer 1) before the manuscript be ready for publication.

I encourage you to carefully consider these aspects while preparing the revised version of your submission.

Kind regards

Marco Milella

Reviewers' comments:

Reviewer's Responses to Questions

**Comments to the Author**

1. Is the manuscript technically sound, and do the data support the conclusions?

Reviewer #1: Partly

Reviewer #2: Yes

2. Has the statistical analysis been performed appropriately and rigorously? 

Reviewer #1: No

Reviewer #2: Yes

3. Have the authors made all data underlying the findings in their manuscript fully available?

Reviewer #1: Yes

Reviewer #2: Yes

4. Is the manuscript presented in an intelligible fashion and written in standard English?

Reviewer #1: Yes

Reviewer #2: Yes

5. Review Comments to the Author

Reviewer #1: Review of the manuscript entitled:

Human mobility on Cancun Island during the 1 Late Postclassic: intra- and inter-site

demographic interactions - Residential mobility on Prehispanic Cancun Island, Mexico

by A. Cucina et al., submitted to PONE

This is an interesting study that applies Sr isotope signatures of human tooth enamel and bones from two archaeological sites on Cancun island with the aim to assess the presence of non-local people on the East Coast of the Yucatan Peninsula during the Late Postclassic (AD 1200-1540). The very peculiar circumstances that make pinpointing of non-locals to these sites lies in the fact that the local bioavailable baseline, dominated by (sub-) modern marine Sr and by Sr derived from extensive limestone platforms dominating the lowlands of the Yucatan peninsula, is very tightly constrained in its range of 87Sr/86Sr values. The Sr isotope analyses of 50 individuals are complemented with the mortuary distribution, and together with other information on dietary habit, funerary patterns, burial location etc., seven individuals were deemed non-locals that apparently did not receive different treatment and had like been integrated into the local community.

The manuscript is well written and logically built up, and some major issues/drawbacks and challenges with applying Sr isotope provenancing using enamel and bones are brought forward and mentioned in the introductory chapter, more precisely in an own chapter “CONCEPTS OF 87SR/86SR ANALYSIS”.

The abstract is precise and informs the readership in short about the major outcome of the study. The same is valid for the conclusions, which are defined and clearly listed.

Being myself not an archeologist, I cannot judge the validity of the archeological background, but in my opinion the authors give a fair and complete summary of the current stage of knowledge, also in the larger framework of what is known from other sites on Yucatan.

There are two issues I have which the authors will have to comment on, discuss and correct-implement in their revised version.

The first one concerns the definition of the local baseline which the author do not clearly indicate in terms of a statistical reference band. The authors, in the technical chapter, state that “To distinguish non-local individuals, the local 87Sr/86Sr range for Isla Cancun was established using limited faunal baseline data (i.e., archaeological opossum [Didelphis sp.] 87Sr/86Sr = 0.7092) and supplemented by previously published values from Mesoamerica (e.g., [31] [32] [45]).”

What exactly is this baseline in 87Sr/86Sr space? It looks to me that the authors chose to make a cut-off at around 87Sr/86Sr ~ 0.70901 (they call it plateau in their Figure 5), which they support by the faunal value they measured on the one opossum). This value is very close to the modern marine signature, and it is doubtful that this cut-off pays respect to the likely influence of very nearby Tertiary limestone dominated bioavailable Sr (for example on drinking water), also in the past. I would strongly suggest the authors broaden their baseline range and incorporate very local signatures of Sr published not so long ago by L.M. Hernandez-Terrones et al. (2021), Continental Shelf Research 212, 104293 (if I am correct, the authors are not citing this work in the submitted manuscript…). This paper summarizes a larger number of environmental 87Sr/86Sr values from the immediate Cancun area, also form the coast near mainland, and not only form the lagoon. I have calculated the average 87Sr/86Sr value of 0.70884 +/- 0.00072 (2s) from the data in this paper and strongly advocate using this range as a conservative and representative range of bioavailable e Sr instead. If implemented, and also indicated in Figure 5, then it becomes clear that the number of non-locals decrease, and locals then are not those that lived on the lagoon only but leaves ample space to also include individuals that lived along the mainland coast just around Cancun where the Tertiary carbonates are prevalent. I could imagine a scenario where individuals could easily move from the mainland coast to the logon and back and that the term “local” would still be applicable under such circumstances.

The second issue I would like to point out and where I would propose the authors come in a bit in more detail in their discussion of the data, is the fact that in the instances where tooth enamel and bone form the same individual were analyzed, it becomes apparent that the bones nearly consistently exhibit a marine Sr isotope signal which in some cases differs strongly form the respective tooth enamel with lower 87Sr/86Sr values. As hinted by the authors vaguely, this to me is a clear sign of diagenetic contamination of the original bone signatures, a fact well known by respective research community. I sternly encourage the authors to discuss this outcome in more detail, also with respect to using such data for the provenance aspects. Also here, it would be very informative if the authors added a realistic baseline range in their Figure 6.

All in all, I enjoyed reading through this manuscript and I recommend publication with moderate revisions along the critical points mentioned above.

11.4.2023 Robert Frei

Reviewer #2: This is a well researched and written paper, and it was such a joy to read your work! Overall, I think that this contribution is vital to bio(archaeological) research - I have a few suggestions/considerations that I think would be helpful to discuss.

A) For Table 3, you present the descriptive statistics of the isotope ratios from San Miguelito and El Rey - Making a box plot graph based on this tables data will also show the slight differences between your groups and adds to your argument about differences/similarities. I would think this would add additional context to the 2 towns/1 town discussion you had as well.

B) Would it be possible to include a map with the structures, and the local/non-local and see if there is a mortuary pattern based on the structure? For example, Burial 26 (5 years, potential non local) and Burial 2 (adult, local) come from structure 1 - so I wonder if you could make a bigger deal about the implications of this data, based on spatial analysis to illuminate other potential social structures?

6. PLOS authors have the option to publish the peer review history of their article (what does this mean?). If published, this will include your full peer review and any attached files.

Reviewer #1: **Yes: **Robert Frei

Reviewer #2: No

---

## [Author Response · Author response to Decision Letter 0]

6 Jul 2023

REPLY TO REVIEWERS

R.DONE

R. TABLES 1 AND 2 REPORT OUR COMPLETE 87SR/86SR DATASET INCLUDING BOTH HUMAN AND BASELINE VALUES. IT WOULD THEREFORE BE REDUNDANT TO ALSO INCLUDE THE VALUES AS SUPPLEMENTARY INFORMATION FILES. WE HAVE CLARIFIED THIS IN OUR DATA AVAILABILITY STATEMENT. 

R: SINCE ALL THE INFORMATION ALREADY APPEARS IN THE OPEN-ACCESS MANUSCRIPT, WE CONSIDER IT USELESS TO ADD THE VERY SAME INFORMATION IN A REPOSITORY.

R: DONE

5. We note that Figures 1-4 in your submission contain [map/satellite] images which may be copyrighted. All PLOS content is published under the Creative Commons Attribution License (CC BY 4.0), which means that the manuscript, images, and Supporting Information files will be freely available online, and any third party is permitted to access, download, copy, distribute, and use these materials in any way, even commercially, with proper attribution. For these reasons, we cannot publish previously copyrighted maps or satellite images created using proprietary data, such as Google software (Google Maps, Street View, and Earth). For more information, see our copyright guidelines: http://journals.plos.org/plosone/s/licenses-and-copyright.

R. FOR FIGURES 1-3, WE HAVE OBTAINED THE PERMISSION FROM THE PERSON WHO MADE THE DRAWINGS. FIGURE 4 HAS BEEN REDRAWN BY THE SENIOR AUTHOR SO NOT TO INCUR IN COPYRIGHT ISSUES. WE ADD FIGURES 7 AND 8 AND WE HAVE ALSO THE PERMISSION FROM THE PERSON WHO MADE THOSE. WE GET THE PERMISSION USING THE CC BY 4.0 LICENSE FOR THESE 5 FIGURES (1 -3 AND 7 AND 8 FIGURES). WE HAVE NOTICE THE FIGURE 1 IS NOT REPRINTED FROM ANY PLACE. IT IS A ORIGINAL FIGURE ELABORATE ON DEMAND OF THE AUTHORS USING A TEMPLATE OF INEGI, THAT IT IS A GOVERNMENTAL INSTITUTION IN MEXICO AND IT IS FREE ACCESS FIGURES 2 AND 3 AND 7 AND 8 ARE DRAWINGS FROM ARCHAEOLOGICAL SITES THAT WERE MADE FOR INSTITUTIONAL PROPOSE AND ARE FREE.

R: DONE

Additional Editor Comments:

We now received the reports from both reviewers.

As you will see, they agree about the scientific interest and overall quality of your work.

They do however recommend that some minor issues be addressed (this especially refers to Reviewer 1) before the manuscript be ready for publication.

I encourage you to carefully consider these aspects while preparing the revised version of your submission.

Kind regards

Marco Milella

Reviewers' comments:

Reviewer's Responses to Questions

Comments to the Author

1. Is the manuscript technically sound, and do the data support the conclusions?

Reviewer #1: Partly

Reviewer #2: Yes

2. Has the statistical analysis been performed appropriately and rigorously?

Reviewer #1: No

Reviewer #2: Yes

3. Have the authors made all data underlying the findings in their manuscript fully available?

Reviewer #1: Yes

Reviewer #2: Yes

4. Is the manuscript presented in an intelligible fashion and written in standard English?

Reviewer #1: Yes

Reviewer #2: Yes

5. Review Comments to the Author

Reviewer #1: Review of the manuscript entitled:

Human mobility on Cancun Island during the 1 Late Postclassic: intra- and inter-site

demographic interactions - Residential mobility on Prehispanic Cancun Island, Mexico

by A. Cucina et al., submitted to PONE

This is an interesting study that applies Sr isotope signatures of human tooth enamel and bones from two archaeological sites on Cancun island with the aim to assess the presence of non-local people on the East Coast of the Yucatan Peninsula during the Late Postclassic (AD 1200-1540). The very peculiar circumstances that make pinpointing of non-locals to these sites lies in the fact that the local bioavailable baseline, dominated by (sub-) modern marine Sr and by Sr derived from extensive limestone platforms dominating the lowlands of the Yucatan peninsula, is very tightly constrained in its range of 87Sr/86Sr values. The Sr isotope analyses of 50 individuals are complemented with the mortuary distribution, and together with other information on dietary habit, funerary patterns, burial location etc., seven individuals were deemed non-locals that apparently did not receive different treatment and had like been integrated into the local community.

The manuscript is well written and logically built up, and some major issues/drawbacks and challenges with applying Sr isotope provenancing using enamel and bones are brought forward and mentioned in the introductory chapter, more precisely in an own chapter “CONCEPTS OF 87SR/86SR ANALYSIS”.

The abstract is precise and informs the readership in short about the major outcome of the study. The same is valid for the conclusions, which are defined and clearly listed.

Being myself not an archeologist, I cannot judge the validity of the archeological background, but in my opinion the authors give a fair and complete summary of the current stage of knowledge, also in the larger framework of what is known from other sites on Yucatan. 

R: WE ARE GLAD THAT DR FREI FOUND THE MS INTERESTING AND WELL STRUCTURED, AND WE APPRECIATE THE USEFUL COMMENTS THAT TRIGGER AN INTERESTING DISCUSSION AROUND THE ISSUE OF WHO IS SUPPOSED TO BE CONSIDERED “LOCAL”. 

There are two issues I have which the authors will have to comment on, discuss and correct-implement in their revised version.

The first one concerns the definition of the local baseline which the author do not clearly indicate in terms of a statistical reference band. The authors, in the technical chapter, state that “To distinguish non-local individuals, the local 87Sr/86Sr range for Isla Cancun was established using limited faunal baseline data (i.e., archaeological opossum [Didelphis sp.] 87Sr/86Sr = 0.7092) and supplemented by previously published values from Mesoamerica (e.g., [31] [32] [45]).”

What exactly is this baseline in 87Sr/86Sr space? It looks to me that the authors chose to make a cut-off at around 87Sr/86Sr ~ 0.70901 (they call it plateau in their Figure 5), which they support by the faunal value they measured on the one opossum). This value is very close to the modern marine signature, and it is doubtful that this cut-off pays respect to the likely influence of very nearby Tertiary limestone dominated bioavailable Sr (for example on drinking water), also in the past. I would strongly suggest the authors broaden their baseline range and incorporate very local signatures of Sr published not so long ago by L.M. Hernandez-Terrones et al. (2021), Continental Shelf Research 212, 104293 (if I am correct, the authors are not citing this work in the submitted manuscript…). This paper summarizes a larger number of environmental 87Sr/86Sr values from the immediate Cancun area, also form the coast near mainland, and not only form the lagoon. I have calculated the average 87Sr/86Sr value of 0.70884 +/- 0.00072 (2s) from the data in this paper and strongly advocate using this range as a conservative and representative range of bioavailable e Sr instead. If implemented, and also indicated in Figure 5, then it becomes clear that the number of non-locals decrease, and locals then are not those that lived on the lagoon only but leaves ample space to also include individuals that lived along the mainland coast just around Cancun where the Tertiary carbonates are prevalent. I could imagine a scenario where individuals could easily move from the mainland coast to the logon and back and that the term “local” would still be applicable under such circumstances.

R: WE DO APPRECIATE DR. FREI’S COMMENTS, WHICH GIVE US A HINT TO IMPROVE THE MANUSCRIPT AND CONTRIBUTE TO SOME EXTENT TO AN INTERESTING ACADEMIC DISCUSSION ABOUT WHO IS LOCAL AND WHO IS NOT BASED ON CLOSE SR (AND OXYGEN) VALUES. THERE IS STILL SO MUCH TO DISCOVER ABOUT ISOTOPIC VALUES IN EACH REGION, AND IT IS TRUE THAT MOBILITY WAS INTENSE ALSO IN PREHISPANIC TIMES. WE ARE AWARE THAT BASELINE DATA TO ESTABLISH “LOCALS” VERSUS “NON-LOCALS” IS A COMPLEX ISSUE. WE ARE ALSO VERY GRATEFUL TO DR. FREI TO REFER TO US THE PAPER BY HERNÁNDEZ-TERRONES ET AL. (2021), WHICH WE CONSIDER USEFUL TO SUPPORT OUR HYPOTHESIS.

WE CROSS-CHECKED THE WATER SAMPLES ANALYZED BY HERNÁNDEZ-TERRONES ET AL. (2021) WITH THE GEOGRAPHICAL LOCATION THEY PROVIDED, BASED ON GOOGLE EARTH. ALL THE SAMPLES FROM CANCUN/PUERTO MORELOS WERE COLLECTED IN AN AREA RANGING BETWEEN 20.84295 TO 21.09336 (LATITUDE), AND 86.81569 AND 86.91689 (LONGITUDE), WHICH ENCOMPASSES INLAND TERRITORIES THAT ARE BETWEEN 16 AND 22 MILES DISTANT FROM THE COASTAL SHORES WHERE CANCUN ISLAND IS LOCATED. THOSE PLACES SHOW A WIDE RANGE OF SR RATIOS; HOWEVER, NO WATER SOURCE HAS BEEN SAMPLED CLOSE TO THE COAST, SO IT IS DIFFICULT TO KNOW THE EXTENT OF VARIABILITY OF THE POTENTIAL WATER SOURCES ALONG THE COAST. IN OUR PERSPECTIVE, IT IS DIFFICULT TO THINK THAT PEOPLE LIVING ON THE COAST WERE RELYING ON WATER SOURCES LOCATED SO FAR INLAND. INSTEAD, THEY MUST HAVE RELIED ON OTHER NOT-YET-SAMPLED WATER SOURCES, AND AT CANCUN ISLAND THERE ARE MANY UNDERGROUND WATER-STORAGE DEPOSITS (CHULTUNES).

GIVEN THE LACK OF WATER/PLANT/GEOLOGICAL SOURCES FOR THE EAST COAST, THE 0.7091 OVERALL VALUES THAT WE CONSIDER AS REPRESENTATIVE OF THE LOCAL PEOPLE ARE THOSE THAT HAVE BEEN CONSISTENTLY REPORTED IN THE MAJORITY OF THE HUMAN CASES (AND ANIMAL SAMPLES) BOTH AT ISLA CANCUN, AT EL MECO, LOCATED JUST NORTH OF CANCUN, AND TULUM. IN FACT, THE AVERAGE VALUE FOR THOSE SUPPOSEDLY LOCAL PEOPLE AT CANCUN ISLAND AND EL MECO (SEE ORTEGA ET AL. 2019) IS 0.70916, WHICH IS WELL ABOVE FROM THE SUGGESTED LOWER LIMIT OF 0.7088.

WE HAVE REPLACED ONE FIGURE (AS REQUESTED BY REVIEWER N. 2), SHOWING THE RANGE OF DISTRIBUTION PER SITE. IN THIS PERSPECTIVE, IT IS INTERESTING TO NOTE THAT, WITH ONLY ONE EXCEPTION OF AN INDIVIDUAL WHOSE SR VALUE IS 0.7088 (THAT HAS BEEN CONSIDERED AS NON-LOCAL ALSO BASED ON HIS OXYGEN VALUE), NO OTHER PEOPLE PRESENTED SR RATIOS BELOW 0.7090. IF THE LOWER LOCAL VALUE WERE, INDEED, 0.7088 (ASSUMING THE LOGICAL MOVEMENT TO AND FROM THE COAST), THEN WE SHOULD EXPECT A MUCH LARGER NUMBER OF PEOPLE WITH SR RATIOS BETWEEN 0.7088 AND 0.7091. INSTEAD, THIS DOES NOT HAPPEN; ONLY ONE INDIVIDUAL - THE ABOVEMENTIONED 0.7088 FOLK - AND TWO INDIVIDUALS WHOSE SR RATIO OF 0.70903 IS SUGGESTIVE AS POTENTIALLY NON-LOCAL, FALL BETWEEN THE 0.7088 AND 0.7091 RANGE, OUT OF 44 (OUT OF 50) MORE PEOPLE FROM THE COAST WITH VALUES ABOVE THE 0.7091 THRESHOLDS (PLUS 13 FROM EL MECO, AND 3 FROM TULUIM – ORTEGA MUÑOZ ET AL. 2019). THEREFORE, WE TRULY THINK THAT THE LOCAL SR RATIO “ALONG” THE COAST (I.E., PEOPLE WHO WERE BORN ON THE COAST, AND NOT 16 TO 22 MILES INLAND), IS AROUND 0.7091. RESTING ON DR. FREI’S COMMENTS, THE QUESTION THAT WE ADDRESS IS: WHICH PEOPLE SHOULD BE CONSIDERED AS “LOCALS” IN CONTRAST WITH “NON-LOCALS”? WHICH IS THE MINIMUM DISTANCE FROM THE COASTAL SHORE OF THE EAST COAST FOR A PERSON TO BE CONSIDERED AS A LOCAL? OUR POSITION IS THAT WE DO NOT THINK THAT PEOPLE WHO POTENTIALLY GREW UP SOME CA. 20 MILES INLAND SHOULD BE CONSIDERED AS “LOCALS” TO THE EAST COAST’ SHORES. AND, BASED ON THE DATASET AVAILABLE, THERE ARE SO FEW PEOPLE WITH SR RATIOS BELOW 0.7090 TO CONSIDER THE 0.7088 VALUE AS THE MINIMUM FOR THE LOCAL RANGE. 

ON THE OTHER HAND, WE THINK, THOUGH WE TREAT THEM WITH CAUTION (AND WE MAKE IT CLEAR IN THE TEXT) THAT THOSE TWO PEOPLE WITH VALUES CLOSE TO THE 0.7090, WHOM WE CONSIDER AS POTENTIALLY NON-LOCALS OF THE EAST COAST’ SHORES, ORIGINATED FROM PLACES NEARBY CANCUN ISLAND. 

The second issue I would like to point out and where I would propose the authors come in a bit in more detail in their discussion of the data, is the fact that in the instances where tooth enamel and bone form the same individual were analyzed, it becomes apparent that the bones nearly consistently exhibit a marine Sr isotope signal which in some cases differs strongly form the respective tooth enamel with lower 87Sr/86Sr values. As hinted by the authors vaguely, this to me is a clear sign of diagenetic contamination of the original bone signatures, a fact well known by respective research community. I sternly encourage the authors to discuss this outcome in more detail, also with respect to using such data for the provenance aspects. Also here, it would be very informative if the authors added a realistic baseline range in their Figure 6.

R: DR. FREI IS PERFECTLY RIGHT WHEN ADDRESSING THE DIAGENETIC ISSUE IN HUMAN BONE. AS SUGGESTED, WE HAVE ADJUSTED OUR INTERPRETATION TO IT. UNFORTUNATELY, WE DO NOT HAVE MUCH MORE INFORMATION ON BONE SR RATIOS TO PROVIDE A MORE DETAILED DISCUSSION SINCE THE METHODOLOGICAL ISSUE OF RECOVERING AND CONFIRMING NON-DIAGENETIC 87SR/86SR FROM BONE APATITE REMAINS UNDER DEBATE (E.G., SEE WATHEN, C.A., ISAKSSON, S. & LIDÉN, K. ON THE ROAD AGAIN—A REVIEW OF PRETREATMENT METHODS FOR THE DECONTAMINATION OF SKELETAL MATERIALS FOR STRONTIUM ISOTOPIC AND CONCENTRATION ANALYSIS. ARCHAEOL ANTHROPOL SCI 14, 45 (2022). HTTPS://DOI.ORG/10.1007/S12520-022-01517-2

All in all, I enjoyed reading through this manuscript and I recommend publication with moderate revisions along the critical points mentioned above.

R: AS SAID, WE REALLY APPRECIATE DR. FREI’S USEFUL COMMENTS, AND WE ARE GRATEFUL FOR THAT.

11.4.2023 Robert Frei

Reviewer #2: This is a well researched and written paper, and it was such a joy to read your work! Overall, I think that this contribution is vital to bio(archaeological) research - I have a few suggestions/considerations that I think would be helpful to discuss.

A) For Table 3, you present the descriptive statistics of the isotope ratios from San Miguelito and El Rey - Making a box plot graph based on this tables data will also show the slight differences between your groups and adds to your argument about differences/similarities. I would think this would add additional context to the 2 towns/1 town discussion you had as well.

R: FIGURE 5 HAS BEEN REDRAWN COMPLETELY, AND NOW IT INCLUDES ALSO A BOX-PLOT (FIG. 5B). A SENTENCE HAS BEEN ADDED TO THE DISCUSSION TO REFER TO THE TWO SITES AS BEING ONE OR TWO SEPARATE POLITICAL ENTITIES.

B) Would it be possible to include a map with the structures, and the local/non-local and see if there is a mortuary pattern based on the structure? For example, Burial 26 (5 years, potential non local) and Burial 2 (adult, local) come from structure 1 - so I wonder if you could make a bigger deal about the implications of this data, based on spatial analysis to illuminate other potential social structures?

R: WE INCLUDE FIGURES 7 AND 8 THAT INCLUDE THE LOCATION OF BURIALS AND THEIR PROVENANCE. DISCUSSION HAS BEEN IMPROVED BASED ON IT.

---

## [Decision Letter · Decision Letter 1]

24 Jul 2023

PONE-D-23-08900R1Human mobility on Cancun Island during the Late Postclassic: intra- and inter-site demographic interactionsPLOS ONE

Dear Dr. Ortega-Muñoz,

Thank you for submitting your manuscript to PLOS ONE. After careful consideration, we feel that it has merit but does not fully meet PLOS ONE’s publication criteria as it currently stands. Therefore, we invite you to submit a revised version of the manuscript that addresses the points raised during the review process.

ACADEMIC EDITOR:Dear Dr. Ortega-Muñoz

Thanks for taking the time to carefully revise your manuscript and to seriouscly consider both reviewers' comments.

Your contribution is now almost ready for publication. However, I strongly suggest you to edit Figures 5 and 6:

Figure 5A: It is not quite clear to me what you are actually showing in it from the relative legend. What are the 2 axes representing? Could you color somehow differently the different sites? Also, it may be beneficial to explicitly indicate in the plot the outliers burial id (rather than isotopic values: these you already have in the tables).

Figure 5B (I am confused by the numbers indicating the outliers: maybe this is a typo and by mistake you indicated the row numbers in SPSS or other variable?).

Figure 6: this is an unusual way to graphically report isotopic values. I suggest you to either plot bone vs enamel values (in this case, residuals from perfect correlation indicate isotopic differences, and their relative direction), or in any case present these data as points (x axis would be individual number, y axis strontium values, each point a sample and 2 different markers for bone and enamel).

In your graphs (especially figures 5B and 6) please use the standard notation for strontium isotopic ratios (87Sr/86Sr).

Kind regards

Marco Milella

We look forward to receiving your revised manuscript.

Kind regards,

Marco Milella

Academic Editor

PLOS ONE

Journal Requirements:

Reviewers' comments:

Reviewer's Responses to Questions

**Comments to the Author**

1. If the authors have adequately addressed your comments raised in a previous round of review and you feel that this manuscript is now acceptable for publication, you may indicate that here to bypass the “Comments to the Author” section, enter your conflict of interest statement in the “Confidential to Editor” section, and submit your "Accept" recommendation.

Reviewer #1: All comments have been addressed

2. Is the manuscript technically sound, and do the data support the conclusions?

Reviewer #1: Partly

3. Has the statistical analysis been performed appropriately and rigorously? 

Reviewer #1: Yes

4. Have the authors made all data underlying the findings in their manuscript fully available?

Reviewer #1: Yes

5. Is the manuscript presented in an intelligible fashion and written in standard English?

Reviewer #1: Yes

6. Review Comments to the Author

Reviewer #1: Dear authors,

you have satisfactorily addressed my two main questions regarding the definition of the baseline and the diagenetic issue, respectively. Please correctly state the baseline range!! you deem appropriate on line 393 after the "+". The added discussion of the Hernandez data and the problem of diagenesis is now at least there, although I am personally still not convinced entirely about the definition of the baseline range. surely some local baseline data are lacking, and maybe this should be added in the text as well! Please also be aware that the Waten paper discusses how bones that that were treated for preservation in respective museums can be cleaned from glue to reveal useful bioavailable Sr signatures, these authors do not discuss on how to recover primary Sr isotope signatures from in situ diagenetic contamination which your bones clearly show. Maybe is can be taken positively, to support your point, that the local Sr signature is actually preserved in the diagenetically alter bones instead?

All in all, good work with some caveats that might be addressed in the future, as you now state in your manuscript.

Robert Frei

7. PLOS authors have the option to publish the peer review history of their article (what does this mean?). If published, this will include your full peer review and any attached files.

Reviewer #1: **Yes: **Robert Frei

---

## [Author Response · Author response to Decision Letter 1]

27 Aug 2023

Figure 5A: It is not quite clear to me what you are actually showing in it from the relative legend. What are the 2 axes representing? Could you color somehow differently the different sites? Also, it may be beneficial to explicitly indicate in the plot the outliers burial id (rather than isotopic values: these you already have in the tables).

R. We change the Figure 5A to bar graph, because, the authors are agreeing that is an easier way to show and comprehend the strontium stable isotopic ratios, with the axe-y with the values of 87Sr/86Sr. We highlight in gray color the non-locals, and We separate San Miguelito from El Rey

Figure 5B (I am confused by the numbers indicating the outliers: maybe this is a typo and by mistake you indicated the row numbers in SPSS or other variable?).

R. Yes it was a mistake We had done again the Figure, with the indicating burial ID.

Figure 6: this is an unusual way to graphically report isotopic values. I suggest you to either plot bone vs enamel values (in this case, residuals from perfect correlation indicate isotopic differences, and their relative direction), or in any case present these data as points (x axis would be individual number, y axis strontium values, each point a sample and 2 different markers for bone and enamel).

R. Yes, Figure 6 has been redone following the suggestion

In your graphs (especially figures 5B and 6) please use the standard notation for strontium isotopic ratios (87Sr/86Sr).

R. DONE

Reviewers' comments:

Reviewer's Responses to Questions

Comments to the Author

Reviewer #1: Dear authors,

you have satisfactorily addressed my two main questions regarding the definition of the baseline and the diagenetic issue, respectively. Please correctly state the baseline range!! you deem appropriate on line 393 after the "+". 

R. Done. It was I mistake of our part. 

The added discussion of the Hernandez data and the problem of diagenesis is now at least there, although I am personally still not convinced entirely about the definition of the baseline range. surely some local baseline data are lacking, and maybe this should be added in the text as well! 

R. We are completely in agreement with you. More research in this perspective about new data to reenforce the baseline in coastal and inland areas of the peninsula of Yucatan, Mexico, is needed. Mainly to evaluate the diagenesis process in bones and teeth in human and faunal of the region.

Please also be aware that the Waten paper discusses how bones that that were treated for preservation in respective museums can be cleaned from glue to reveal useful bioavailable Sr signatures, these authors do not discuss on how to recover primary Sr isotope signatures from in situ diagenetic contamination which your bones clearly show. Maybe is can be taken positively, to support your point, that the local Sr signature is actually preserved in the diagenetically alter bones instead?

R. The samples process to Sr analysis were not treated with any chemical products (glue, mowital, or others). The cleaning process was with distilled water to remove the dirt or soil matrix. We discard diagenetic contamination both from the place that these individuals were buried because the isotopic signature is diverse, but with small range. For example, Price and colleagues (2020: 592) show that the values of 87Sr/86Sr of bones samples of sailors of the voyages of Columbus to America, buried in the Santo Domingo Island, are generally homogenous (mean: 07092+- 0.00002), and they consider that these samples are contaminated by local strontium isotopes. Our samples have a mean of 0.70918+- 0.00003, (min=0.709119; max= 0.709214) and from fauna is 0.70920. Then, the diagenetic contamination could not be discarded. We have pointed out this aspect about contamination in the manuscript and its limitation t implies, at least for the bony remains.

---

## [Editor Report · Decision Letter 2]

11 Sep 2023

Human mobility on Cancun Island during the Late Postclassic: intra- and inter-site demographic interactions

PONE-D-23-08900R2

Dear Dr. Ortega-Muñoz,

We’re pleased to inform you that your manuscript has been judged scientifically suitable for publication and will be formally accepted for publication once it meets all outstanding technical requirements.

Kind regards,

Marco Milella

Academic Editor

PLOS ONE
---

## [Editor Report · Acceptance letter]

27 Sep 2023

PONE-D-23-08900R2 

Human mobility on Cancun Island during the Late Postclassic: intra- and inter-site demographic interactions 

Dear Dr. Ortega-Muñoz:

I'm pleased to inform you that your manuscript has been deemed suitable for publication in PLOS ONE. Congratulations! Your manuscript is now with our production department. 

Kind regards, 

on behalf of

Dr. Marco Milella 

Academic Editor

PLOS ONE